# Where Did I Come From? Origin Attribution of AI-Generated Images

**Zhenting Wang**\*
Rutgers University
zhenting.wang@rutgers.edu

**Chen Chen**
Sony AI
ChenA.Chen@sony.com

**Yi Zeng**
Virginia Tech
yizeng@vt.edu

**Lingjuan Lyu**†
Sony AI
Lingjuan.Lv@sony.com

**Shiqing Ma**
University of Massachusetts Amherst
shiqingma@umass.edu

## Abstract

Image generation techniques have been gaining increasing attention recently, but concerns have been raised about the potential misuse and intellectual property (IP) infringement associated with image generation models. It is, therefore, necessary to analyze the origin of images by inferring if a specific image was generated by a particular model, i.e., origin attribution. Existing methods only focus on specific types of generative models and require additional procedures during the training phase or generation phase. This makes them unsuitable for pre-trained models that lack these specific operations and may impair generation quality. To address this problem, we first develop an alteration-free and model-agnostic origin attribution method via reverse-engineering on image generation models, i.e., inverting the input of a particular model for a specific image. Given a particular model, we first analyze the differences in the hardness of reverse-engineering tasks for generated samples of the given model and other images. Based on our analysis, we then propose a method that utilizes the reconstruction loss of reverse-engineering to infer the origin. Our proposed method effectively distinguishes between generated images of a specific generative model and other images, i.e., images generated by other models and real images.

## 1 Introduction

In recent years, there has been a rapid evolution in image generation techniques. With the advances in visual generative models, images can now be easily created with high quality and diversity [1–4]. There are three important milestones in the field of image generation and manipulation, i.e., Generative Adversarial Networks (GAN) [5], Variational AutoEncoders (VAE) [6], and diffusion models [7]. Various image generation models are built based on these three models [8–13] to make the AI-generated images more realistic.

With its wide adoption, the security and privacy of image generation models becomes critical [14–19]. One severe and important issue is the potential misuse and intellectual property (IP) infringement of image generation models [16, 18]. Users may generate malicious images containing inappropriate or biased content using these models and distribute them online. Furthermore, trained models may be used without authorization, violating the model owner's intellectual property. For example, malicious users may steal the model's parameters file and use it for commercial purposes. Others may create AI-generated images and falsely claim them as their own artwork (e.g., photos and paintings) to

---

\*Work partially done during Zhenting Wang's internship at Sony AI.
†Corresponding Author

37th Conference on Neural Information Processing Systems (NeurIPS 2023).

gain recognition, which also violates the model's IP. Therefore, it is essential to track the origin of AI-generated images. The origin attribution problem is to identify whether a specific image is generated by a particular model. As shown in Fig. 1, assuming we have a model $\mathcal{M}_1$ and its generated images, the origin attribution algorithm's objective is to flag an image as belonging to model $\mathcal{M}_1$ if it was generated by that model. On the other hand, the algorithm should consider the image as non-belonging if it was created by other models (e.g., $\mathcal{M}_2$ in Fig. 1) or if it is a real image.

One existing way to infer the source of specific images is image watermarking. It works by embedding ownership information in carrier images to verify the owner's identity and authenticity [20–23]. The image watermarking-based method requires an additional modification to the generation results in a post-hoc manner, which may impair the generation quality. Also, it might not necessarily reflect the use of a particular model in the process of generating an image, which can reduce its credibility as evidence in lawsuits. Furthermore, the watermark can also be stolen, and malicious users can engage in criminal activities and disguise their identities us-

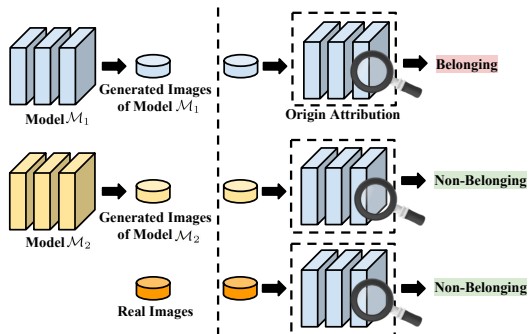

Fig. 1: Illustration for origin attribution problem. The origin attribution algorithm aims to judging whether the given images belong to a particular model, i.e., Model $\mathcal{M}_1$.

ing the stolen watermark. Another approach to identifying the source models of the generated samples [24–27] is injecting fingerprinting into the models (e.g., modifying the model architecture) and training a supervised classifier to detect the fingerprints presented in the image. While their goal is similar to ours, these methods have several limitations. Firstly, they require extra operations during the model training phase, and they cannot be applied on pre-trained models without additional operations, such as modifying the model architecture [26, 27]. Secondly, since these methods modify the training or inference process of generative models, the model's generation performance may be affected. Thirdly, these previous studies mainly focus on a particular kind of generative model, i.e., GAN [5]. In contrast, our goal is to develop an origin attribution approach for different types of generative models, including diffusion models [4, 3] (model-agnostic), without requiring any extra operations in the training phase and image generation phase (alteration-free). We summarize the differences between our method and existing methods in Table 1.

In this paper, we propose a method for origin attribution that is based on the input reverse-engineering task on generative models (i.e., inverting the input of a particular generative model for a specific image). The intuition behind our method is that the reverse-engineering task is easier for belonging images than non-belonging images. Therefore, we design our method

Table 1: Summary of the differences between our method and existing methods.

| Method | Training-phase Alteration-free | Generation-phase Alteration-free | Model-agnostic |
|---|---|---|---|
| Image Watermark [20–23] | ✔ | ✘ | ✔ |
| Classifier-based [24–27] | ✘ | ✔ | ✘ |
| Ours | ✔ | ✔ | ✔ |

based on the differences in the reconstruction loss for the reverse-engineering between generated images of a given model and other images. The origin attribution method we propose starts by using the model to generate a set of images and calculate the reconstruction loss on these generated images. To eliminate the influence of the inherent complexities of the examined images, we also calibrate the reconstruction loss by considering the hardness of the reverse-engineering on any other model that has strong generation ability, but different architectures and training data. Afterwards, the algorithm computes the calibrated reconstruction loss for the examined image and uses statistical hypothesis testing to determine if the reconstruction loss falls within the distribution of reconstruction losses observed in the generated images. This allows us to distinguish images generated by the given model from other images in a model-agnostic and alteration-free manner. Based on our design, we implemented a prototype called RONAN (**R**everse-engineering-based **O**rigi**N** **A**ttributio**N**) in PyTorch and evaluated it on three different types of generative models (i.e., unconditional model, class-to-image model, and text-to-image model) including various GANs [1, 10, 28–30], VAEs [6], and diffusion models such as latest Consistency Model [4] and Stable Diffusion [3]. Results demonstrate our

method is effective for the "alteration-free and model-agnostic origin attribution" task. On average, RONAN achieves 96.07% of true positive rate with a false positive rate around 5.00%.

Our contributions are summered as follows: ① We introduce a new task called "alteration-free and model-agnostic origin attribution", which entails determining whether a specific image is generated by a particular model without requiring any additional operations during the training and generation phases. ② To accomplish this task, we analyze the differences in the reconstruction loss for reverse-engineering between the generated images of a given model and other images. Based on our analysis, we design a novel method that involves conducting input reverse-engineering and checking whether the reconstruction loss of the examined sample falls within the distribution of reconstruction losses observed in the generated images. ③ We evaluate our method on various different image generation models. The results show that our method effectively distinguishes images generated by the given model from other images in a model-agnostic and alteration-free manner. Our code can be found in https://github.com/ZhentingWang/RONAN.

## 2 Related Work

**Detection of AI-Generated Contents.** Detecting AI-generated content has become extremely important with the growing concerns about the misuse of AIGC technology [31]. The detection of AI-generated content is a binary classification problem that involves distinguishing generated samples from real ones. In the CV field, existing research has found that visually imperceptible but machine-distinguishable patterns in generated images, such as noise patterns [16, 32], frequency signals [33–36] and texture representation [37] can be used as the clues of AI-generated images. Researchers also proposed methods to detect sentences generated by generative NLP models such as ChatGPT [38–41]. Although these methods achieve promising performance for distinguishing AI-generated content and real content, they cannot infer if a specific content is generated by a given generative model, which is a novel but more challenging task and is the main focus of this paper.

**Tracking Origin of Generated Images.** There are several ways to track the source of the generated images. Image watermarking that pastes ownership information in carrier images [20–23] can be adapted to discern whether a specific sample is from a specific source. Watermarks can take the form of specific signals within the images, such as frequency domain signals [22] or display-camera transformations [42]. However, it requires an additional modification to the generation results in a post-hoc manner, and it does not necessarily reflect whether the image was generated by a particular model when the judges use it as the evidence (different models can use the same watermark). Another way is to inject fingerprints into the model during training and train a supervised classifier on it to discern whether an image is from a fingerprinted GAN model [24–27]. For example, Yu et al. [27] modify the architecture of the convolutional filter to embed the fingerprint, and they train the generator alongside a fingerprinting classifier capable of identifying the fingerprint and its corresponding source GAN models. It requires a modified model architecture, altered training process, and an additional procedure to train a source classifier. There are several existing methods [43–45] track the origin of the generated images by exploiting the input reconstruction. However, the practicality of these approaches is limited due to their reliance on strong assumptions (see § 5.5 for more details).

## 3 Problem Formulation

We focus on serving as an inspector to infer *if a specific sample is generated by a particular model in an alteration-free and model-agnostic manner*. To the best of our knowledge, this paper is the first work focusing on this problem. To facilitate our discussion, we first define the belonging and non-belonging of the generative models.

**Definition 3.1** (Belonging of Generative Models). Given a generative model $\mathcal{M} : \mathcal{I} \mapsto \mathcal{X}_{\mathcal{M}}$ where $\mathcal{I}$ is the input space and $\mathcal{X}_{\mathcal{M}}$ is the output space. A sample $x$ is a **belonging** of model $\mathcal{M}$ if and only if $x \in \mathcal{X}_{\mathcal{M}}$. We call a sample $x$ is a **non-belonging** if $x \notin \mathcal{X}_{\mathcal{M}}$.

**Inspector's Goal.** Given a sample $x$ and a generative model $\mathcal{M} : \mathcal{I} \mapsto \mathcal{X}_{\mathcal{M}}$ where $\mathcal{I}$ is the input space and $\mathcal{X}_{\mathcal{M}}$ is the output space of $\mathcal{M}$, the inspector's goal is to infer if a given image $x$ is a belonging of $\mathcal{M}$. Formally, the goal can be written as constructing an inference algorithm $\mathcal{B} : (\mathcal{M}, x) \mapsto \{0, 1\}$ that receives a sample $x$ and a model $\mathcal{M}$ as the input, and returns the inference result (i.e., 0 denotes belongings, and 1 denotes non-belongings). The algorithm $\mathcal{B}$ can distinguish not only the belongings of the given model $\mathcal{M}$ and that of the other models (e.g., trained on different training data, or have different model architectures), but also the belongings of $\mathcal{M}$ and the natural samples that are not generated by AI. The inspector also aims to achieve the following goals:

*Alteration-free:* The algorithm $\mathcal{B}$ does not require any extra operations/modifications in the training phase and image generation phase.

*Model-agnostic:* The algorithm $\mathcal{B}$ can be applied to different types of image generative models with different architectures.

**Inspector's Capability.** The inspector has white-box access to the provided model $\mathcal{M}$, thus the inspector can get the intermediate output and calculate the gradient of the models. In addition, the inspector cannot control the development and training process of the provided models. Note that the white-box access to the inspected model is practical especially in the scenarios where the inspector is the owner of the inspected models.

**Real-world Application.** The inspection algorithm can be widely used in various scenarios where it is necessary to verify the authenticity of generated images. We provide three examples as follows:

*Copyright protection of image generation models:* A copyright is a kind of intellectual property (IP) that provides its owner the exclusive right to perform a creative work [46]. In this scenario, a party suspects that a specific image may have been generated by their generative model without authorization, such as if a malicious user has stolen the model and used it to generate images. The party can then request an inspector to use our proposed method to infer if the doubtful image was indeed generated by their particular model, and the resulting inference can be used as a reference in a lawsuit. It is important to clarify that infering an image as a belonging of a model does not imply the IP of this image is totally belonging to this model. In fact, determining the ownership of the IP related to the generated images remains an unresolved challenge in the field of law. This complexity arises due to the involvement of multiple entities (such as contributors of training data, model trainers, input/prompt providers, and the models themselves) throughout the image generation process. The infering results of our origin attribution method can serve as a valuable reference for addressing IP protection concerns, instead of a definitive conclusion.

*Tracing the source of maliciously generated images:* Assume a user creates malicious images containing inappropriate or biased content and distributes them on the internet. The cyber police can utilize our proposed method to infer if the image was generated by a model belonging to a specific user. The resulting inference can be used as a reference for criminal evidence in lawsuits.

*Detecting AI-powered plagiarism:* Our method can also be used for detecting AI-powered plagiarism. For example, imagine a scenario where an individual generates AI-created images (e.g., using Midjourney) and dishonestly presents them as their own original artwork (e.g., photographs and paintings) to gain recognition and reputation. In such cases, the model owner (e.g., Midjourney's owner) may suspect that the image is generated using their model (e.g., Midjourney). Our proposed method can then be employed to uncover such instances of AI-powered plagiarism.

# 4 Method

Our method is built on the input reverse-engineering for image generation models. In this section, we start by formulating the input reverse-engineering task, followed by an analysis of the disparities in reconstruction loss between images generated by a particular model and those from other sources. We then proceed to present a detailed algorithm for our method.

## 4.1 Reverse-engineering

Recent researches [47, 17] demonstrate that the input of the modern image generative models is reconstructable by using the generated images. Here, we view the reverse-engineering as an optimization problem. Formally, it can be defined as follows:

**Definition 4.1** (Input Reverse-engineering). Given a generative model $\mathcal{M} : \mathcal{I} \mapsto \mathcal{X}_\mathcal{M}$, and an image $\boldsymbol{x}$, an input reverse-engineering task is optimizing the input $\boldsymbol{i}$ to make the corresponding output image from the model $\mathcal{M}(\boldsymbol{i})$ as close as possible to the given image $\boldsymbol{x}$.

The input reverse-engineering task is performed by a **reverse-engineering algorithm** $\mathcal{A} : (\mathcal{M}, \boldsymbol{x}) \mapsto R$, which can be written as:

$$\boldsymbol{i}^\star = \arg\min_{\boldsymbol{i}} \mathcal{L}\left(\mathcal{M}(\boldsymbol{i}), \boldsymbol{x}\right), \quad \mathcal{A}(\mathcal{M}, \boldsymbol{x}) = \mathcal{L}\left(\mathcal{M}(\boldsymbol{i}^\star), \boldsymbol{x}\right) \tag{1}$$

where $\mathcal{L}$ is a metric to measure the distance between different images, and $\boldsymbol{i}^\star$ is the reverse-engineered input. The given value for the reverse-engineering algorithm $\mathcal{A}$ is a specific image $\boldsymbol{x}$ and a particular model $\mathcal{M}$. The returned value of the algorithm $\mathcal{A}$ is the distance between the given image and its

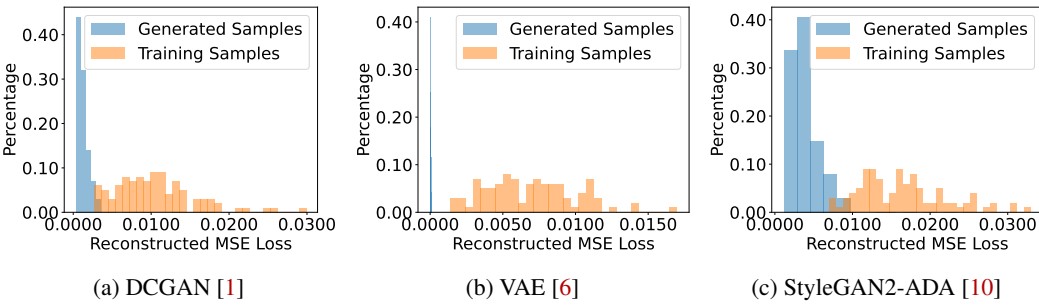

Fig. 2: Reconstruction loss distributions for belonging images and real images.

reverse-engineered version, i.e., $\mathcal{L}\left(\mathcal{M}(\boldsymbol{i}^\star), \boldsymbol{x}\right)$, which is called as the **reconstruction loss**. We use the reconstruction loss to measure the hardness of the input reverse-engineering task.

Based on the definitions and formulations, we have the following theorem:

**Theorem 4.2.** *Given a generative model $\mathcal{M} : \mathcal{I} \mapsto \mathcal{X}_{\mathcal{M}}$, and a reverse-engineering algorithm $\mathcal{A}$, if the model is deterministic (i.e., it produces the same output given the same input) and the reverse-engineering algorithm is perfect (i.e., it can find the global minimum of the reconstruction loss for the reverse-engineering), then for any $\boldsymbol{x} \in \mathcal{X}_{\mathcal{M}}$ (belonging) and $\boldsymbol{x}' \notin \mathcal{X}_{\mathcal{M}}$ (non-belonging) we have $\mathcal{A}(\mathcal{M}, \boldsymbol{x}') > \mathcal{A}(\mathcal{M}, \boldsymbol{x})$.*

The proof for Theorem 4.2 can be found in the Appendix A. The theorem demonstrates that the reconstruction loss of the images generated by a specific model will be lower than that of images that are not generated by the model. The theorem also establishes that the distribution of reconstruction loss values for belonging and non-belonging images is perfectly separable. Thus, we can use a threshold value to separate the belonging images and non-belonging images. In the real world, many image generation models incorporate random noises into their image generation procedures to enhance the variety of images they produce. However, these models can also be deemed deterministic since we can regard all the random noises utilized in the generation procedure as parts of the inputs. On the other hand, in reality, the reverse-engineering algorithm may get stuck at a local minimum, and it is hard to guarantee the achievement of the global minimum. This is where the formula $\mathbb{P}(\mathcal{A}(\mathcal{M}, \boldsymbol{x}') > \mathcal{A}(\mathcal{M}, \boldsymbol{x})) \geq \lambda$ becomes relevant as it serves as a relaxation for Theorem 4.2, explaining the practical scenario. In this formula, $\lambda$ (e.g., $90\%$) acts as a separability level for distinguishing between the two distributions: belonging images and non-belonging images.

To investigate the practical scenario, we conduct experiments on the CIFAR-10 [48] dataset using DCGAN [1], VAE [6], and StyleGAN2-ADA [10] models. The results are depicted in Fig. 2, where the x-axis represents the reconstruction loss measured by the MSE (Mean Squared Error) [49] metric, and the y-axis indicates the percentage of images whose reconstruction loss value corresponds to the corresponding value on the x-axis. We use blue color to denote 100 generated images of the given model and orange to represent 100 real images randomly sampled from the training data of the model. The results indicate that the reconstruction losses of the generated images (belongings) and those not generated by this model (non-belongings) can be distinguished.

### 4.2 Calibration

Different images have different inherent complexities [50–52]. Some images may be harder to reverse-engineer due to their higher complexity (e.g., containing more objects, colors, and details). In that case, the reconstruction loss will also be influenced by the inherent complexities of the examined images. To increase the separability level of belonging images and others, we disentangle the influence of the inherent complexities and the belonging status by measuring the complexity of the image and use it to calibrate the reconstruction loss. By default, we use the reconstruction loss on a reference image generation model $\mathcal{M}_r$ that is trained on a different dataset as the measurement of the complexity, i.e., $\text{Complexity}(\boldsymbol{x}) = \mathcal{A}(\mathcal{M}_r, \boldsymbol{x})$ (we use the Consistency model [4] pre-trained on ImageNet dataset [53] as the reference model by default). We also discuss other implements to measure the complexity of the inspected images in Appendix F. The calibrated reconstruction loss $\mathcal{A}'(\mathcal{M}, \boldsymbol{x})$ is defined as follows:

$$\mathcal{A}'(\mathcal{M}, \boldsymbol{x}) = \frac{\mathcal{A}(\mathcal{M}, \boldsymbol{x})}{\text{Complexity}(\boldsymbol{x})} \tag{2}$$

---
**Algorithm 1** Origin Attribution
---
**Input:**   Model: $\mathcal{M}$, Examined Data: $x$
**Output:**   Inference Results: Belonging or Non-belonging
 1: **function** INFERENCE($\mathcal{M}, x$)
 2:     ▷ Obtaining Belonging Distribution (Offline)
 3:     $\mu, \sigma, N = \text{BelongingDistribution}(\mathcal{M})$
 4:     ▷ Reverse-engineering
 5:     $\mathcal{A}'(\mathcal{M}, x) \leftarrow$ Calibrated Reconstruction Loss [Eq. 2]
 6:     ▷ Determining Belonging
 7:     InferenceResults = HypothesisTesting($\mathcal{A}'(\mathcal{M}, x), \mu, \sigma, N$)[Eq. 3]
 8:     **return** InferenceResults
---

### 4.3   Belonging Inference via Hypothesis Testing

We use Grubbs' Hypothesis Testing [54] to infer if a specific sample $x$ is a belonging of the particular given model $\mathcal{M}$. We have a null hypothesis $H_0 : x$ *is a non-belonging of* $\mathcal{M}$, and the alternative hypothesis $H_1 : x$ *is a belonging of* $\mathcal{M}$. The null hypothesis $H_0$ is rejected (i.e., the alternative hypothesis $H_1$ is accepted) if the following inequality (Eq. 3) holds:

$$\frac{\mathcal{A}'(\mathcal{M}, x) - \mu}{\sigma} < \frac{(N-1)}{\sqrt{N}} \sqrt{\frac{\left(t_{\alpha/N, N-2}\right)^2}{N - 2 + \left(t_{\alpha/N, N-2}\right)^2}} \tag{3}$$

Here, $\mu$ and $\sigma$ are the mean value and standard deviation for the calibrated reconstruction loss on belonging samples of model $\mathcal{M}$. Since model $\mathcal{M}$ is given to the inspector, the inspector can calculate $\mu$ and $\sigma$ by using $\mathcal{M}$ to generate multiple images with randomly sampled inputs. $N$ is the number of generated belonging images. $\mathcal{A}'(\mathcal{M}, x)$ is the calibrated reconstruction loss of the examined image $x$. $t_{\alpha/N, N-2}$ is the critical value of the $t$ distribution with $N - 2$ degrees of freedom and a significance level of $\alpha/N$, where $\alpha$ is the significance level of the hypothesis testing (i.e., 0.05 by default in this paper). The critical value of the $t$ distribution (i.e., $t_{\alpha/N, N-2}$) can be computed using the cumulative distribution function (See Appendix B for more details).

### 4.4   Algorithm

We propose Algorithm 1 to determine if a specific sample belongs to a given model. The input of Algorithm 1 is the examined data $x$ and the given model $\mathcal{M}$. The output of this algorithm is the inference results, i.e., belonging or non-belonging. In line 3, we use the given model to generate $N$ (i.e., 100 by default in this paper) images with randomly sampled inputs and calculate the mean value ($\mu$) and standard deviation ($\sigma$) for the calibrated reconstruction loss on the generated belonging samples. This step can be done offline, i.e., it only needs to be performed once for each model. In line 5, we calculate the calibrated reconstruction loss of the examined image (Eq. 2), the reconstruction loss is computed via gradient descent optimizer (Adam [55] by default in this paper). In line 7, we determine if the examined image $x$ belongs to the model $\mathcal{M}$ or not by conducting the Grubbs' Hypothesis Testing [54] (§ 4.3). The given image is flagged as a belonging of the given model if the corresponding hypothesis is accepted.

## 5   Experiments and Results

In this section, we first introduce the setup of the experiments (§ 5.1). We evaluate the effectiveness of RONAN (§ 5.2) and provide a case study on Stable Diffusion v2 model [3] (§ 5.3). We then conduct ablation studies in § 5.4, and discuss the comparison to existing reconstruction based attribution methods in § 5.5. The discussion about the efficiency and robustness against image editing can be found in the Appendix.

### 5.1   Setup

Our method is implemented with Python 3.8 and PyTorch 1.11. We conducted all experiments on a Ubuntu 20.04 server equipped with six Quadro RTX 6000 GPUs.

**Models.** Eleven different models are included in the experiments: DCGAN [1], VAE [6], StyleGAN2-ADA [10], DDIM [2], DDPM [7], TransGAN [56], StyleGAN XL [30], Consistency Diffusion

Table 2: Detailed results on distinguishing belonging images and real images.

| Model Type | Model | Training Dataset | Belongings vs Training Data | | | | | Belongings vs Unseen Data | | | | |
|---|---|---|---|---|---|---|---|---|---|---|---|---|
| | | | TP | FP | FN | TN | Acc | TP | FP | FN | TN | Acc |
| Unconditional | DCGAN | CIFAR-10 | 96 | 0 | 4 | 100 | 98.0% | 95 | 0 | 5 | 100 | 97.5% |
| | VAE | CIFAR-10 | 95 | 0 | 5 | 100 | 97.5% | 96 | 0 | 4 | 100 | 98.0% |
| Class-conditional | StyleGAN2-ADA | CIFAR-10 | 96 | 2 | 4 | 98 | 97.0% | 95 | 0 | 5 | 100 | 97.5% |
| | Consistency Model | ImageNet | 96 | 24 | 4 | 76 | 86.0% | 96 | 11 | 4 | 89 | 92.5% |
| Text-conditional | ControlGAN | CUB-200-2011 | 95 | 10 | 5 | 90 | 92.5% | 96 | 8 | 4 | 92 | 94.0% |

Table 3: Results for distinguishing belonging images and images generated by other models with different architectures. Here, Model $\mathcal{M}_1$ is the examined model, Model $\mathcal{M}_2$ is the other model that has same training data but different architectures.

| Training Dataset | Model $\mathcal{M}_1$ | Model $\mathcal{M}_2$ | TP | FP | FN | TN | Acc |
|---|---|---|---|---|---|---|---|
| CIFAR-10 | DCGAN | VAE | 96 | 0 | 4 | 100 | 98.0% |
| | | StyleGAN2ADA | 96 | 0 | 4 | 100 | 98.0% |
| | | DDIM | 97 | 0 | 3 | 100 | 98.5% |
| | | DDPM | 96 | 0 | 4 | 100 | 98.0% |
| | | TransGAN | 97 | 0 | 3 | 100 | 98.5% |
| | VAE | DCGAN | 97 | 0 | 3 | 100 | 98.5% |
| | | StyleGAN2ADA | 95 | 0 | 5 | 100 | 97.5% |
| | | DDIM | 96 | 0 | 4 | 100 | 98.0% |
| | | DDPM | 96 | 0 | 4 | 100 | 98.0% |
| | | TransGAN | 97 | 0 | 3 | 100 | 98.5% |
| | StyleGAN2ADA | DCGAN | 96 | 2 | 4 | 98 | 97.0% |
| | | VAE | 95 | 1 | 5 | 99 | 97.0% |
| | | DDIM | 96 | 1 | 4 | 99 | 97.5% |
| | | DDPM | 96 | 0 | 4 | 100 | 98.0% |
| | | TransGAN | 95 | 6 | 5 | 94 | 94.5% |
| ImageNet | Consistency Model | StyleGAN XL | 95 | 8 | 5 | 92 | 93.5% |
| | StyleGAN XL | Consistency Model | 96 | 10 | 4 | 90 | 93.0% |
| CUB-200-2011 | ControlGAN | StackGAN-v2 | 96 | 17 | 4 | 83 | 89.5% |
| | StackGAN-v2 | ControlGAN | 96 | 14 | 4 | 86 | 91.0% |

Model [4], Control-GAN [28], StackGAN-v2 [29], and Stable Diffusion v2 [3]. These models are representative image generation models.

**Performance Metrics.** The effectiveness of the method is measured by collecting the detection accuracy (Acc). For a particular model, given a set of belonging images and non-belonging images, the Acc is the ratio between the correctly classified images and all images. We also show a detailed number of True Positives (TP, i.e., correctly detected belongings), False Positives (FP, i.e., non-belongings classified as belongings), False Negatives (FN, i.e., belongings classified as non-belongings) and True Negatives (TN, i.e., correctly classified non-belongings).

## 5.2 Effectiveness

In this section, we evaluate the effectiveness of RONAN from two perspectives: (1) its effectiveness in distinguishing between belongings of a particular model and real images; (2) its effectiveness in differentiating between belongings of a particular model and those generated by other models.

**Distinguishing Belonging Images and Real Images.** To investigate RONAN's effectiveness in distinguishing between belonging images of a particular model and real images, given an image generation model, we start by differentiating between the generated images of the given model and the training data of the model. The investigated models include DCGAN [1], VAE [6], StyleGAN2-ADA [10] trained on the CIFAR-10 [48] dataset, Consistency Model [4] trained on the ImageNet [53] dataset, and ControlGAN [28] trained on the CUB-200-2011 [57] dataset. Among them, DCGAN and VAE are unconditional image generation models. StyleGAN2-ADA and the latest diffusion model Consistency Model, are class-conditional models. In addition to distinguishing belongings and the training data, we also conduct experiments to distinguish belongings from unseen data that has a similar distribution to the training data (i.e., the test data of the dataset). For each case, we evaluate the results on 100 randomly sampled belonging images and 100 randomly sampled non-belonging images. The results are demonstrated in Table 2. As can be observed, the detection accuracies (Acc) are above 85% in all cases. On average, the Acc is 94.2% for distinguishing belongings and training data, and is 95.9% for distinguishing belongings and unseen data. The results demonstrate that

RONAN achieves good performance in distinguishing between belonging images of a particular model and real images.

**Distinguishing Belonging Images and Images Generated by Other Models.** In this section, we study RONAN's effectiveness to distinguish between belonging images of a particular model and the images generated by other models. For a given model $\mathcal{M}_1$, we consider two different types of other models $\mathcal{M}_2$, i.e., the model trained on the same dataset but with different architectures and the model that has the same architecture but is trained on a different dataset.

*Models with Different Architectures.* We first evaluate RONAN's effectiveness in distinguishing between belonging images of a particular model and the images generated by other models with the same training data but different architectures. For the models trained on the CIFAR-10 [48] dataset, the used model architectures are DCGAN [1], VAE [6], StyleGAN2-ADA [10], DDIM [2], DDPM [7], TransGAN [56]. For the Imagenet [53] dataset, the involved models are the latest diffusion model Consistency Model [4] and StyleGAN XL [30]. For the CUB-200-2011 [57] dataset, we use text-to-image models ControlGAN [28] and StackGAN-v2 [29]. To measure the effectiveness of RONAN, we collect its results on 100 randomly sampled belonging images and 100 randomly sampled non-belonging images. The results are shown in Table 3, where Model $\mathcal{M}_1$ denotes the examined model, and Model $\mathcal{M}_2$ represents the other model that has the same training data but a different architecture. The results show that the average detection accuracy (Acc) of RONAN is 96.45%, confirming its good performance for distinguishing between belongings of a given model and the images generated by other models with different architectures.

*Models Trained on Different Datasets.* We also evaluate RONAN's effectiveness in distinguishing between belonging images of a particular model and the images generated by other models with the same model architecture but trained on different datasets. The model used here is the diffusion model Consistency Model [4]. We use a model trained on the ImageNet [53] dataset and a model trained on the LSUN [58] dataset. The results are demonstrated in Table 4, where Model $\mathcal{M}_1$ denotes the examined model, and Model $\mathcal{M}_2$ means the other model that has the same model architecture but is trained on a different dataset. The results show that RONAN can effectively distinguish between belonging images of a particular model and the images generated by other models with the same model architecture but different training data. On average, the detection accuracy of our method is 93.0%. In the Appendix, we also discuss the results when the training data of the

Table 4: Results for distinguishing belonging images and images generated by other models with different training data. Here, Model $\mathcal{M}_1$ is the examined model, Model $\mathcal{M}_2$ is the other model that has same architecture but different training data.

| Trainging dataset of Model $\mathcal{M}_1$ | Trainging dataset of Model $\mathcal{M}_2$ | TP | FP | FN | TN | Acc |
|---|---|---|---|---|---|---|
| ImageNet | LSUN | 95 | 13 | 5 | 87 | 91.0% |
| LSUN | ImageNet | 96 | 6 | 4 | 94 | 95.0% |

model $\mathcal{M}_2$ and that of the model $\mathcal{M}_1$ have overlaps. Empirical results demonstrate that RONAN is still effective even the training data of the examined model and that of the other model are similar (i.e., they have large overlaps).

## 5.3 Case Study on Stable Diffusion v2

In this section, we conduct a case study on the recent Stable Diffusion v2 [3] model. We first randomly collect 20 images of Shiba dogs from the internet and use these images as the non-belonging images. We then use the prompt "A cute Shiba on the grass." and feed it into the Stable Diffusion v2 model to generate 20 belonging images. We apply RONAN on the model, and evaluate its performance in distinguishing the belonging images and non-belonging images. The results show that the detection accuracy of RONAN is 87.5%, with 18 TP, 3 FP, 2 FN, 17 TN. In Fig. 3, we show the visualization of a belonging image and a non-belonging image, as well as their corresponding reverse-engineered images. Note that the non-belonging image and the belonging image have similar inherent complexities (i.e., their reconstructed loss with the MSE metric on the reference model are 0.029 and 0.022, respectively). For the non-belonging image, the reverse-engineered image is more noisy and blurred, while the reverse-engineered image of the belonging image seems nearly identical to the original image. These results show the potential to apply our method on state-of-the-art models such as Stable Diffusion v2. More visualizations and examples can be found in Appendix J.

## 5.4 Ablation Study

In this section, we evaluate the impact of different metrics used in calculating reconstruction loss, and the impact of reconstruction loss calibration.

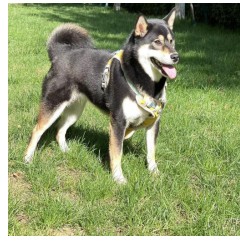 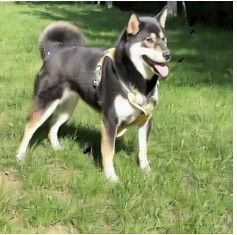 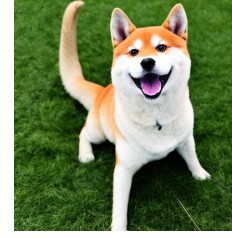 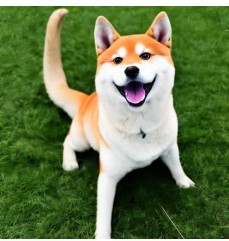

| Non-belonging | Reverse-engineered | | Belonging | Reverse-engineered |

(a) A non-belonging image and its reverse-engineered image for Stable Diffusion v2. The reconstructed loss is 0.0078 with MSE metric.

(b) A belonging image and its reverse-engineered image for Stable Diffusion v2. The reconstructed loss is 0.0005 with MSE metric.

Fig. 3: Visualization of the belonging image and non-belonging image for Stable Diffusion v2 [3], and their corresponding reverse-engineered images. The non-belonging image and the belonging image have similar inherent complexities.

**Different Metrics.** In the reverse-engineering task (Eq. 1), we use a metric $\mathcal{L}$ to measure the distance between different images. By default, we select MSE [49] as the metric. In addition to MSE, we also evaluate the results on other image distance metrics, i.e., MAE (Mean Absolute Error) [59], SSIM (Structural Similarity Index Measure) [60], and LPIPS (Learned Perceptual Image Patch Similarity) [61]. The task is to distinguish between belonging images and real images (i.e., training images of the model here). The model used in this section is the StyleGAN2-ADA [10] trained on CIFAR-10 [48] dataset. The results are shown in Table 5. As we can observe, the detection accuracy (Acc) under MAE, MSE, SSIM, and LPIPS are 94.5%, 97.0%, 85.5%, and 96.0%, respectively. Overall, the MSE metric achieves the highest performance in distinguishing belonging images and real images. Thus, we select MSE as our default metric.

Table 5: Results on different distance metrics.

| Metric | TP | FP | FN | TN | Acc |
|---|---|---|---|---|---|
| MAE | 96 | 7 | 4 | 93 | 94.5% |
| MSE | 96 | 2 | 4 | 98 | 97.0% |
| SSIM | 97 | 26 | 3 | 74 | 85.5% |
| LPIPS | 96 | 4 | 4 | 96 | 96.0% |

**Impacts of Reconstruction Loss Calibration.** To eliminate the influence of images' inherent complexities, we calibrate the reconstruction loss by considering the hardness of the reverse-engineering on a reference model (§ 4.2). To measure the effects of the calibration step, we compare the detailed TP, FP, FN, TN, and Acc for the method with and without the calibration step. We use the Stable Diffusion v2 [3] model and follow the experiment settings described in § 5.3. The results in Table 6 demonstrate the detection accuracy for the method with and without the calibration step are 75.0% and 87.5%, respectively. These results show that the calibration step can effectively improve the performance of RONAN.

Table 6: Effects of reconstruction loss calibration.

| Method | TP | FP | FN | TN | Acc |
|---|---|---|---|---|---|
| w/o Calibration | 17 | 7 | 3 | 13 | 75.0% |
| w/ Calibration | 18 | 3 | 2 | 17 | 87.5% |

### 5.5 Comparison to Existing Reconstruction based Attribution Methods

Albright et al. [43] and Zhang et al. [44] are two existing attribution methods that also apply the reconstruction on the input. In this section, we compare our method to them. Given an inspected image, Albright et al. and Zhang et al. works by enumerating and conducting inversion on all models within a set of suspicious candidate models (referred to as the "candidate set" in this section), and attribute the model with the lowest reconstruction loss as the source of the image. Their methods rely on the assumptions that the inspector can have the white-box access to all models in the candidate set, and the examined image must be generated by one of the models in the candidate set. These assumptions diminish the practicability of their methods, whereas our method does not have such requirements. Thus, our threat model and experiment settings are fundamentally different to theirs. Our paper focus on the problem of determining if a given image is generated by a single given model or not. Consider a scenario where a model owner wants to verify if an image is generated by a model owned by him/her. While our method only needs to conduct the inversion on this specific model. Albright et al. and Zhang et al. need to compare the reconstruction loss of this particular model with a large number of suspicious models. There are several cases where Albright et al. and Zhang et al. are ineffective in addressing this problem: ① In cases where the inspector lacks white-box access

to some of the suspicious models, deriving reconstruction on them and getting inference results becomes infeasible. Notably, more and more state-of-the-art image generation models (e.g., such as Midjourney [62] and DALL-E2 [13]) are close-sourced and they only provide the black-box API to the users. ② Albright et al. and Zhang et al. are prone to making wrong predictions if the real source model is not included in the candidate set. This is attributed to their underlying assumption that the examined image must originate from one of the models within the candidate set. Ensuring the real source model is included within the candidate set is a very hard problem in practice. ③ Equally noteworthy, Albright et al. and Zhang et al. do not work when the inspected images are real images due to their strong assumption (i.e., the examined image must be generated by one of the model in the candidate set). Our method does not have the above problems.

Despite the threat models are different, we also conduct the comparison experiments. We consider the setting for distinguishing the belonging images of the inspected model $\mathcal{M}_1$ and the generated images of other models $\mathcal{M}_2$, and the inspected model here is Stable Diffusion v2. For

Table 7: Comparison to Albright et al. [43] and Zhang et al. [44].

| Model $\mathcal{M}_1$ | Model $\mathcal{M}_2$ | Method | TP | FP | FN | TN | Acc |
|---|---|---|---|---|---|---|---|
| Stable Diffusion v2 | StyleGAN2-ADA | Albright et al. | 100 | 100 | 0 | 0 | 50.0% |
| | | Zhang et al. | 100 | 100 | 0 | 0 | 50.0% |
| | | Ours | 96 | 7 | 4 | 93 | 94.5% |
| | Consistency Model | Albright et al. | 100 | 100 | 0 | 0 | 50.0% |
| | | Zhang et al. | 100 | 100 | 0 | 0 | 50.0% |
| | | Ours | 95 | 10 | 5 | 90 | 92.5% |

our method, we assume the inspector can only access $\mathcal{M}_1$. For Albright et al. and Zhang et al., we assume the inspector can access both $\mathcal{M}_1$ and $\mathcal{M}_2$. The results can be found in Table 7. While the average detection accuracy of Albright et al. and Zhang et al. are only 50%, our method achieves over 90% detection accuracy, meaning that our method outperforms Albright et al. and Zhang et al.

Laszkiewicz et al. [45] is a concurrent work focusing on the single-model attribution problem. Although the method proposed in Laszkiewicz et al. achieves promising performance, it is important to note that it is not model-agnostic. More specifically, it assumes the last layer of the inspected model is invertible. Therefore, it is not applicable to the models that use non-invertible activation functions (such as the commonly-used ReLU function) in the last layer. It also has assumptions on the architectures of the models, making it can not be used for many modern diffusion models [7, 63].

## 6 Discussion

**Limitations.** While our method can achieve origin attribution in a alteration-free and model-agnostic manner, the computation cost might be higher than that of watermark-based methods [20–23] and classifier-based methods [24–27]. More discussion about the efficiency of our method can be found in the Appendix. Speeding up the reverse-engineering will be our future work. This paper focuses on the image generation models. Expanding our method for origin attribution on the model in other fields, (e.g., vedio generation models such as Imagen Video [64], language generation models like ChatGPT [65], and graph generation models [66, 67]) will also be our future work.

**Ethics.** Research on security and privacy of machine learning potentially has ethical concerns [68–78]. This paper proposes a method to safeguard the intellectual property of image generation models and monitor their potential misuse. We believe that our approach will enhance the security of image generation models and be beneficial to generative AI.

## 7 Conclusion

In this paper, we take the first effort to introduce the "alteration-free and model-agnostic origin attribution" task for AI-generated images. Our contributions for accomplishing this task involves defining the reverse-engineering task for generative models and analyzing the disparities in reconstruction loss between generated samples of a given model and other images. Based on our analysis, we devise a novel method for this task by conducting input reverse-engineering and calculating the corresponding reconstruction loss. Experiments conducted on different generative models under various settings demonstrate that our method is effective in tracing the origin of AI-generated images.

## Acknowledgement

We thank the anonymous reviewers for their valuable comments. This research is supported by Sony AI, IARPA TrojAI W911NF-19-S-0012, NSF 2342250 and 2319944. Any opinions, findings, and conclusions expressed in this paper are those of the authors only and do not necessarily reflect the views of any funding agencies.

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

**Roadmap:** In this appendix, we show the proof for Theorem 4.2 (Appendix A), detailed process for calculating critical value of the $t$ distribution used in § 4.3 (Appendix B), more results for distinguishing belonging images and images generated by other models (Appendix C), a thought experiment on VAE (Appendix D), results on different reverse-engineering methods (Appendix E), discussion on different methods for measuring the complexity of the images (Appendix F). RONAN's robustness to editing-based adaptive attack (Appendix G), the discussion for the efficiency of RONAN (Appendix H). Finally, we provide more results and demonstrate more visualizations on Stable Diffusion v2 [3] (Appendix I andAppendix J).

## A  Proof for Theorem 4.2

We start our analysis from ideal generative model and reconstruction algorithm, which we define as deterministic generative model and perfect reconstruction algorithm:

**Definition A.1** (Deterministic Generative Model). Given a generative model $\mathcal{M} : \mathcal{I} \mapsto \mathcal{X}_{\mathcal{M}}$, it is deterministic if it always produce the same output $x \in \mathcal{X}_{\mathcal{M}}$ given the same input $i \in \mathcal{I}$.

**Definition A.2** (Perfect Reverse-engineering Algorithm). Given a reverse-engineering algorithm $\mathcal{A}$, if it is guaranteed that the returned reconstruction loss $l$ is the global minima, then we say $\mathcal{A}$ is a perfect reverse-engineering algorithm.

*Proof.* Assume the given output sample $x$ is generated by input $i$. Since the given model $\mathcal{M}$ is deterministic, we have:

$$x = \mathcal{M}(i) \tag{4}$$

In this case, the distance between the $x$ and $\mathcal{M}(i)$ is 0, i.e., $\mathcal{L}(\mathcal{M}(i), x) = 0$. Based on Theorem 4.1, as $\mathcal{A}$ is a perfect reverse-engineering algorithm, it can find the input that can produce the minimal reconstruction loss. Therefore, we have:

$$\forall x \in \mathcal{X}_{\mathcal{M}}, \mathcal{A}(\mathcal{M}, x) = 0 \tag{5}$$

Similarly, for sample $x' \notin \mathcal{X}_{\mathcal{M}}$. There does not exist an input $i'$ that can produce $x'$, meaning that there does not exist an input $i'$ that have $\mathcal{L}(\mathcal{M}(i'), x') = 0$. Thus, we have:

$$\forall x' \notin \mathcal{X}_{\mathcal{M}}, \mathcal{A}(\mathcal{M}, x') > 0 \tag{6}$$

Finally, we have for any $x \in \mathcal{X}_{\mathcal{M}}$ and $x' \notin \mathcal{X}_{\mathcal{M}}$ we have $\mathcal{A}(\mathcal{M}, x') > \mathcal{A}(\mathcal{M}, x)$.

$\square$

## B  Computing Critical Value of the $t$ Distribution.

In § 4.3, we use the critical value of the $t$ distribution to obtain the results of the hypothesis testing. In this section, we discuss the detailed process for calculating the critical value. For the t-distribution, we have the probability density function:

$$f(t) = \frac{\Gamma\left(\frac{\nu+1}{2}\right)}{\sqrt{\nu\pi}\Gamma\left(\frac{\nu}{2}\right)}\left(1 + \frac{t^2}{\nu}\right)^{-(\nu+1)/2} \tag{7}$$

In Eq. 7, $\nu$ is the number of degrees of freedom and $\Gamma$ is the gamma function. Based on Eq. 7, we have the cumulative distribution function:

$$\mathbb{P}(t < t') = \int_{-\infty}^{t'} f(u)du = 1 - \frac{1}{2}\beta\left(\frac{\nu}{t'^2+\nu}, \frac{\nu}{2}, \frac{1}{2}\right) \tag{8}$$

where $\beta$ denotes the incomplete beta function. Therefore, given a confidence level $\alpha$ and the number of degrees of freedom $\nu$, we have can use Eq. 9 to obtain the value of the critical value $t_{\alpha,\nu}$.

$$\mathbb{P}(t < t_{\alpha,\nu}) = 1 - \alpha \tag{9}$$

Table 9: Results when the inspected model and the other model has the similar architectures but their numbers of layers are different.

| $\mathcal{M}_1$'s Number of Layers | $\mathcal{M}_2$'s Number of Layers | Acc |
|---|---|---|
| 4 | 2 | 97.5% |
| 2 | 4 | 98.0% |

Table 10: Results when the inspected model and the other model has the similar architectures but their number of channels in the first Conv layer are different.

| $\mathcal{M}_1$'s Number of Channels in the first Conv Layer | $\mathcal{M}_2$'s Number of Channels in the first Conv Layer | Acc |
|---|---|---|
| 64 | 48 | 97.5% |
| 48 | 64 | 96.0% |

## C  More Results for Distinguishing Belonging Images and Images Generated by Other Models.

In this section, we provide more results for distinguishing belonging images of the model and the images generated by other models. We first consider distinguishing the belonging images and the images generated by the other models having the same architecture and overlapped training data. Given a model $\mathcal{M}_1$, we focus on the other model $\mathcal{M}_2$ which shares the same architecture as $\mathcal{M}_1$ and has training data that overlaps with $\mathcal{M}_2$ 's training data. The model architecture used here is DCGAN [1]. We trained $\mathcal{M}_1$ on the full CIFAR-10 [48] dataset, while $\mathcal{M}_2$ is trained on the randomly sampled subsets of the CIFAR-10 dataset. The results are presented in Table 8, where the first column indicates the proportion of overlap between the training data of $\mathcal{M}_1$ and $\mathcal{M}_2$. The second column of Table 8 displays the accuracy of RONAN in detecting the differences. Notably, even when 90% of $\mathcal{M}_2$'s training data overlaps with that of $\mathcal{M}_1$, the detection accuracy remains above 95%. These results demonstrate that our method is still effective

Table 8: Results when the inspected model and the other model has the same architecture and their training data has overlaps.

| Overlap Fraction | Acc |
|---|---|
| 50% | 98.5% |
| 60% | 98.0% |
| 70% | 98.5% |
| 80% | 96.0% |
| 90% | 96.5% |

when the training data of the inspected model and that of the other model are similar. We also consider distinguishing the belonging images and the images generated by the other models having the same training data and similar architectures. We use DCGAN [1] here and modifies the number of layers and the number of channels in the first Conv layer in the models to make $\mathcal{M}_1$ and $\mathcal{M}_2$ have similar architectures. Both $\mathcal{M}_1$ and $\mathcal{M}_2$ here are trained on CIFAR-10 [48] dataset. The results can be found in Table 9 and Table 10. Based on these results, our method is effective even the inspected model and other models have the same training dataset and similar architectures. We also provide a summary of the combinatorial setups used for distinguishing belonging images and images generated by other models in Table 11.

## D  A Thought Experiment on VAE.

If a image generation model perfectly fits the training samples, then the images generated by it will be indistinguishable to its training samples (which are real images). In this section, we investigate a question: "Can image generation models perfectly fit the training samples?". To study this question, we conduct a thought experiment on VAE [6]. In detail, we randomly sample 10 images from MNIST [79] dataset, and use VAE [6] with different numbers of neurons in the hidden layer to fit these 10 images. We set the epoch number to 5000 to ensure the training losses that measures the distance between the generated samples and the training samples of the model are converged. The detailed final training losses with different model sizes are shown in Table 12. There are two parts in the total training loss $\mathcal{L}_{\text{total}}$: the reconstruction part $\mathcal{L}_{\text{reconstruction}}$ and the KL-divergence part $\mathcal{L}_{\text{divergence}}$, i.e., $\mathcal{L}_{\text{total}} = \mathcal{L}_{\text{reconstruction}} + \mathcal{L}_{\text{divergence}}$. In Table 12, Detailed values for each part are demonstrated.

Table 11: Summary of combinatorial setups for distinguishing belonging images and images generated by other models.

| Setting | $\mathcal{M}_1$ | | $\mathcal{M}_2$ | |
|---|---|---|---|---|
| | Model | Training Dataset | Model | Training Dataset |
| Different Model Architecture | DCGAN | CIFAR-10 | VAE | CIFAR-10 |
| | DCGAN | CIFAR-10 | StyleGAN2ADA | CIFAR-10 |
| | DCGAN | CIFAR-10 | DDIM | CIFAR-10 |
| | DCGAN | CIFAR-10 | DDPM | CIFAR-10 |
| | DCGAN | CIFAR-10 | TransGAN | CIFAR-10 |
| | VAE | CIFAR-10 | DCGAN | CIFAR-10 |
| | VAE | CIFAR-10 | StyleGAN2ADA | CIFAR-10 |
| | VAE | CIFAR-10 | DDIM | CIFAR-10 |
| | VAE | CIFAR-10 | DDPM | CIFAR-10 |
| | VAE | CIFAR-10 | TransGAN | CIFAR-10 |
| | StyleGAN2ADA | CIFAR-10 | VAE | CIFAR-10 |
| | StyleGAN2ADA | CIFAR-10 | DCGAN | CIFAR-10 |
| | StyleGAN2ADA | CIFAR-10 | DDIM | CIFAR-10 |
| | StyleGAN2ADA | CIFAR-10 | DDPM | CIFAR-10 |
| | StyleGAN2ADA | CIFAR-10 | TransGAN | CIFAR-10 |
| | Consistency Model | ImageNet | StyleGAN XL | ImageNet |
| | StyleGAN XL | ImageNet | Consistency Model | ImageNet |
| | ControlGAN | CUB-200-2011 | StackGAN-v2 | CUB-200-2011 |
| | StackGAN-v2 | CUB-200-2011 | ControlGAN | CUB-200-2011 |
| Similar Model Architecture | DCGAN | CIFAR-10 | DCGAN with different numbers of layers | CIFAR-10 |
| | DCGAN | CIFAR-10 | DCGAN with different numbers of channels | CIFAR-10 |
| Different Training Dataset | Consistency Model | ImageNet | Consistency Model | LSUN |
| | Consistency Model | LSUN | Consistency Model | ImageNet |
| Overlapping Training Dataset | DCGAN | CIFAR-10 | DCGAN | 50% subset of CIFAR-10 |
| | DCGAN | CIFAR-10 | DCGAN | 60% subset of CIFAR-10 |
| | DCGAN | CIFAR-10 | DCGAN | 70% subset of CIFAR-10 |
| | DCGAN | CIFAR-10 | DCGAN | 80% subset of CIFAR-10 |
| | DCGAN | CIFAR-10 | DCGAN | 90% subset of CIFAR-10 |

Table 12: Detailed final training losses with different model sizes in the thought experiments on VAE.

| Num of Neurons in Hidden Layer | $\mathcal{L}_{\text{total}}$ | $\mathcal{L}_{\text{reconstruction}}$ | $\mathcal{L}_{\text{divergence}}$ |
|---|---|---|---|
| 1 | 184.24 | 183.93 | 0.31 |
| 5 | 105.27 | 99.68 | 5.59 |
| 10 | 77.20 | 69.07 | 8.13 |
| 50 | 53.52 | 47.55 | 5.97 |
| 100 | 53.23 | 47.99 | 5.24 |
| 500 | 53.04 | 48.21 | 4.83 |
| 1000 | 53.81 | 48.93 | 4.88 |

With the increases in the model sizes, the final total training losses reduce at first. However, when the model is large enough, the final total training loss becomes stable at the region from 53.00-54.00. The results show that even the VAEs with enough large sizes can not fit all pixels perfectly. We also conduct the reverse-engineering of the training samples on the trained VAE with 1000 neurons in the hidden layer. The results show that we can not reconstruct the exact training samples.

As pointed out by existing works [33, 80, 81], morden generative models (e.g., VAEs, GANs, and Diffusion models) including the state-of-the-art models such as Stable Diffusion [3] and DALL-E [13] will leave forensics traces in the frequency spaces of the generated images. This is caused by indispensable operations used in modern generative models (e.g., Upsampling operations and Convolutional Layers). Also, perfectly fitting the training samples means finding the global optimum in the optimization process. However, the essential gradient descent optimizers used in modern generative models (e.g., SGD and Adam) typically can not get the global optimum. Based on the above analysis and empirical results, at least we can conclude that the probabilities that the real images belong to the output space of generative models are very low.

# E Discussion on Different Reverse-engineering Methods

In this section, we discuss the details of the reverse-engineering process on different models and the effects of different reverse-engineering methods. The details of the reverse-engineering for different

Table 13: Details of reverse-engineering on different models.

| Model | Details of Reverse-engineering |
|---|---|
| DCGAN | Using gradient decent to optimize the input in the noise space. |
| VAE | Using gradient decent to optimize the input in the noise space. |
| StyleGAN2-ADA | Using gradient decent to optimize the input in the noise space. |
| Consistency Model | Using gradient Decent to optimize the input in the noise space. |
| StyleGAN XL | Using gradient decent to optimize the input in the noise space. |
| ControlGAN | Using gradient decent to optimize the feature in the intermediate feature space before the $G_3$ layer. |
| StackGAN-v2 | Using gradient decent to optimize the feature in the intermediate feature space before the $G_2$ layer. |
| Stable Diffusion 2 | Using gradient decent to optimize the feature in the intermediate feature space before the decoder. |

Table 14: Results under different inversion methods for StyleGANs.

| Model | Space | Acc |
|---|---|---|
| StyleGAN2-ADA | Z space | 97.0% |
| | $\mathcal{W}$+ space | 96.0% |
| StyleGAN XL | Z space | 93.0% |
| | $\mathcal{W}$+ space | 93.5% |

Table 15: Results under different inversion methods for Stable Diffusion.

| Inversion Method | Acc |
|---|---|
| Default | 87.5% |
| DDIM Inversion | 55.5% |
| Parmar et al. | 62.5% |

models are summarized in Table 13. We also investigate the effects of different reverse-engineering methods. For StyleGAN2-ADA [10] and StyleGAN-XL [30], we use the random noise space (i.e., Z space) to reconstruct the images by default. We also conducted experiments on using intermediate latent space (i.e., $\mathcal{W}$+ space) to reconstruct the images for distinguishing belonging images and images generated by other models (see setting for Table 3). The results on StyleGAN2-ADA with the CIFAR-10 [48] dataset and StyleGAN XL with ImageNet dataset are shown in Table 14. As can be observed, using the $\mathcal{W}$+ space has similar accuracy to using Z space, meaning that our method is not sensitive to the selection of the input spaces used for optimization.

We also conduct experiments for different reverse-engineering methods designed for diffusion models (i.e., DDIM inversion [2], Parmar et al. [82] and our method) on the Stable Diffusion v2 [3] model with the setting described in § 5.3. The results are shown in Table 15. DDIM inversion [2] is an inversion approach performing the reverse of the DDIM sampling. It is based on the assumption that the ODE process can be reversed in limited of small steps. It only has unsatisfying inversion performance on the conditional diffusion models (e.g., Stable Diffusion) because it will magnify the accumulated error in the inversion process (since it ignores the classifier-free guidance in the diffusion process). As can be seen in the above table, the accuracy of using DDIM inversion is only 55.5%. Parmar et al. [82] improve the DDIM inversion by using an approximated prompt as the conditional guidance in the inversion process. The approximated prompts are generated by a caption model (i.e., BLIP). This method's inversion quality is dependent on the captions used. Given a generated image of the inspected model, since the caption model can not get the accurate prompt used for generating this image, the inversion using this method is also not accurate, leading that the reconstruction losses of the belonging images and non-belonging images are not highly separable (the accuracy is 62.5%). Compared to DDIM inversion [2] and Parmar et al. [82], our method achieves higher accuracy.

## F  Discussion on Different Methods for Measuring the Complexity of the Images

In this section, we discuss the different methods for measuring the complexity of the images. We first discuss the reference model based method. Here, we conducted experiments that using different models as reference models. The inspected model here is the StyleGAN2-ADA [10] model trained on CIFAR-10 [48] dataset and the setting here is identical to that used in the Table 2 (belongings vs training data). The results can be found in Table 16. As can be observed, using

Table 16: Results on using different models as reference models.

| Reference Model | Acc |
|---|---|
| Consistency Model | 97.0% |
| StyleGAN XL | 95.0% |
| Stable Diffusion | 96.0% |

different reference models yields similar results, meaning that our method is not sensitive to the selection of the reference models. The measurement of the image complexity can also be implemented

Table 17: Results under adaptive attack with instagram filter editing.

| TP | FP | FN | TN | Acc |
|---|---|---|---|---|
| 96 | 15 | 4 | 85 | 90.5% |

Table 18: Results under adaptive attack with box blur filter editing.

| Box Size | Acc | SSIM |
|---|---|---|
| 1 | 92.5% | 0.8920 |
| 2 | 83.0% | 0.7446 |
| 3 | 58.0% | 0.5174 |
| 4 | 53.5% | 0.3530 |

by calculating the 2D entropy of the image [83]. Using this entropy-based method as the image complexity measurement also yields 97.0% final detection accuracy in our setting.

# G Adaptive Attack

In this section, we evaluate the robustness of RONAN against the adaptive attack where the malicious user is aware of it and try to bypass the inspection of RONAN. For example, when the malicious user use a specific model to generate an image, he/she can make a slight modification on the image to bypass the inspection of the origin attribution algorithm. We consider the image editing as the adaptive attack because it can preserve most of the information in the original image while changing the image. To investigate if RONAN is robust to the image-editing-based adaptive attack, we use the _1977 instagram filter[3] and the box blur filter to conduct results. The model used here is the DCGAN [1] model trained on the CIFAR-10 [48] dataset. The results are shown in Table 17 and Table 18. For _1977 instagram filter, we can see that the detection accuracy of RONAN is still above 90% even under the adaptive attack. For box blur filter, we show the results under different box sizes of the filter. Besides the detection accuracy (Acc) of our method, we also demonstrate the Structural Similarity Index (i.e., SSIM [60]) between the original images and the edited images, which can measure the similarity between them. A higher SSIM value means the edited images are more similar to the original images. As can be observed, our method remains effective under relatively small box sizes. As the box sizes expand, however, the detection accuracy of our method diminishes. This outcome is understandable and acceptable, as it corresponds to a rapid reduction in the Structural Similarity Index (SSIM) between the edited images and their unaltered counterparts. When employing larger box sizes, it is conceivable that an adaptive attacker might find ways to elude our method's scrutiny, yet this comes at the cost of substantially compromising the quality of the edited images. Consequently, our method maintains its effectiveness even in the face of adaptive attacks that seek to maintain the quality of the edited images.

# H Efficiency

In this section, we discuss the efficiency of RONAN. To study the efficiency, we measure the runtime of our method on different models (i.e., DCGAN [1], VAE [6], StyleGAN2-ADA [10], Consistency Model [4], and Stable Diffusion v2 [3]). For each model, we run 5 times on one Quadro RTX 6000 GPU and collect the average runtime and its standard deviation. The results can be found in Table 19. Our method can be accelerated by using mixed precision training [84]. Further approach for speeding up the input reverse-engineering process will be our future work.

Table 19: Runtime on different models.

| Model | Runtime (s) |
|---|---|
| DCGAN | $16.53 \pm 0.35$ |
| VAE | $51.63 \pm 0.84$ |
| StyleGAN2-ADA | $54.44 \pm 0.79$ |
| Consistency Model | $153.65 \pm 2.02$ |
| Stable Diffusion | $605.45 \pm 3.26$ |

# I More Results on Stable Diffusion

In this section, we provide more results on the Stable Diffusion. Here, we use the Stable Diffusion v2 model as the inspected model. The scenario considered here is distinguishing 1000 images generated by Stable Diffusion v2 model (the inspected model) and 1000 images generated by DeepFloyd-IF-II-L [4] model. Note that the images here are generated using the prompts randomly sampled from the

---

[3]https://github.com/akiomik/pilgram
[4]https://huggingface.co/DeepFloyd/IF-II-L-v1.0

PromptHero [5] website. The results demonstrate that our method achieves 91.6% detection accuracy, meaning it is effective under this scenario.

## J   More Visualizations on Stable Diffusion

In this section, we provide more visualizations for our case study on Stable Diffusion v2 model [3] (§ 5.3). We show more visualizations for the belonging images and their corresponding reverse-engineered images in Fig. 4. In Fig. 5, we demonstrate more visualizations for the non-belonging images of the Stable Diffusion v2 model, and also show their reverse-engineered images. The detailed reconstruction loss and the calibrated reconstruction loss are reported in Fig. 4 and Fig. 5. The distance metric used here is MSE. As we can observed, the belonging images have much lower calibrated reconstruction loss than the non-belonging images, further demonstrating the effectiveness of our approach.

---

[5]https://prompthero.com/

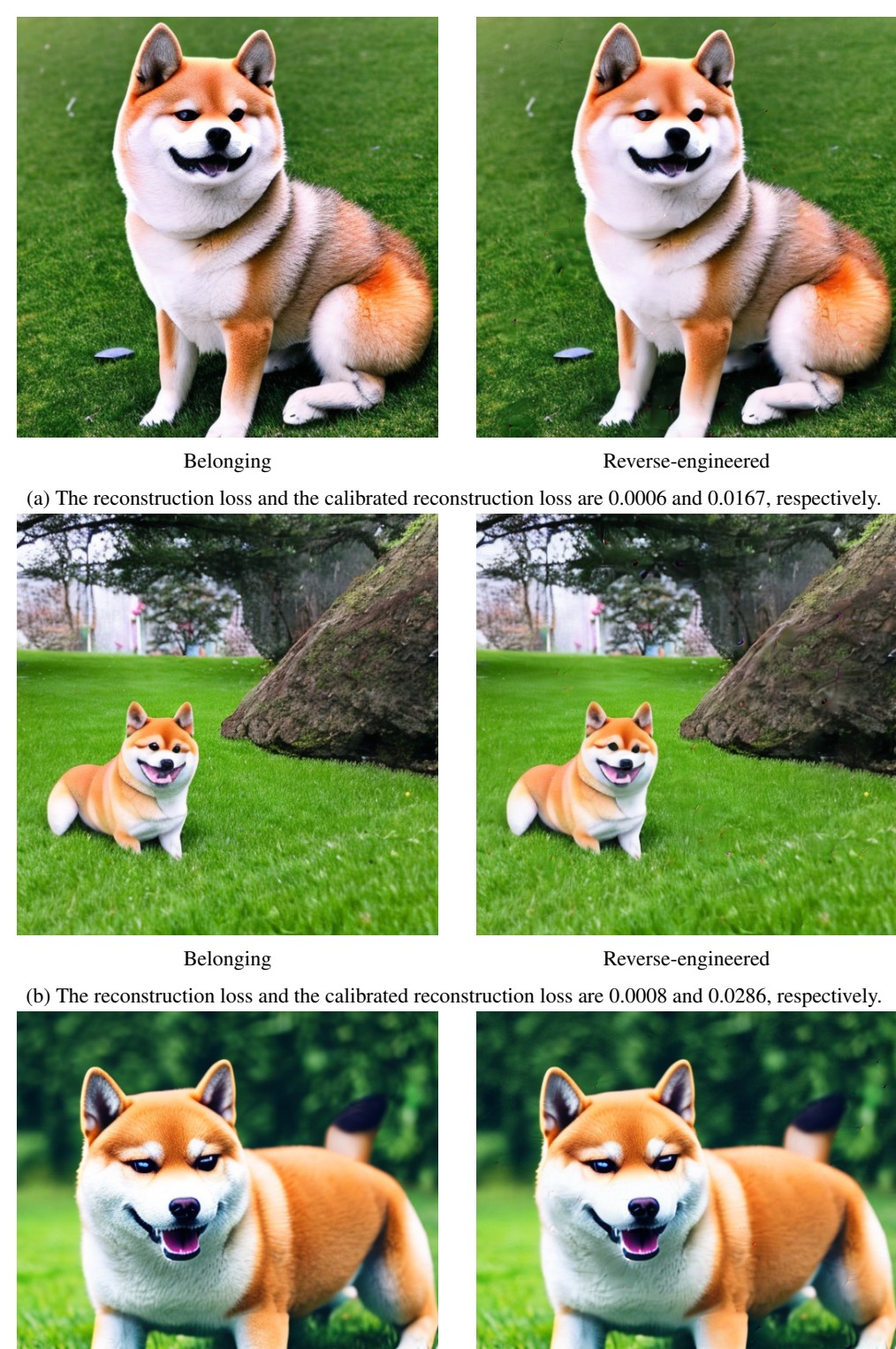

Belonging                    Reverse-engineered

(a) The reconstruction loss and the calibrated reconstruction loss are 0.0006 and 0.0167, respectively.

Belonging                    Reverse-engineered

(b) The reconstruction loss and the calibrated reconstruction loss are 0.0008 and 0.0286, respectively.

Belonging                    Reverse-engineered

(c) The reconstruction loss and the calibrated reconstruction loss are 0.0005 and 0.0161, respectively.

Fig. 4: More visualization of the belonging images for Stable Diffusion v2 [3], and their corresponding reverse-engineered images. The distance metric used here is MSE.

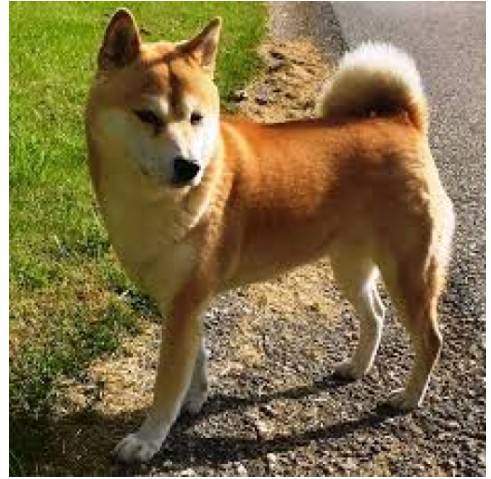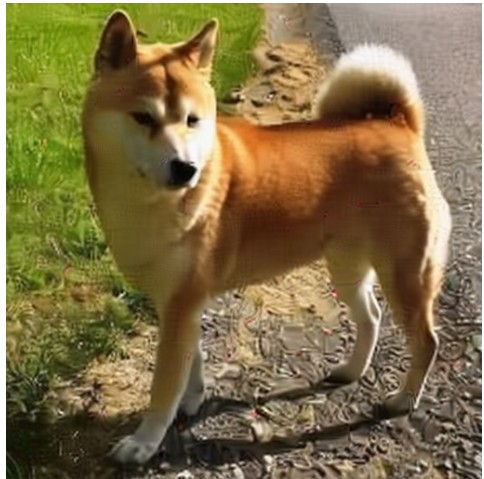

Non-belonging                                    Reverse-engineered

(a) The reconstruction loss and the calibrated reconstruction loss are 0.0038 and 0.1727, respectively.

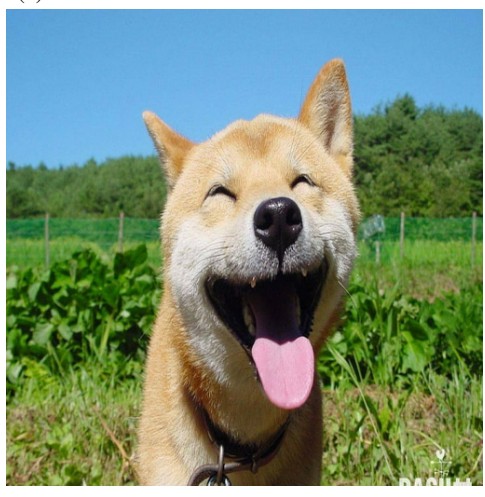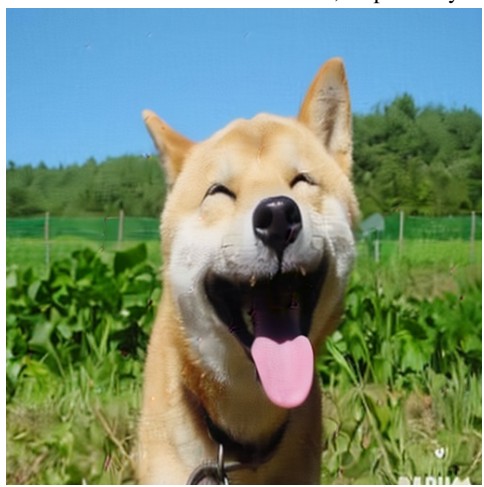

Non-belonging                                    Reverse-engineered

(b) The reconstruction loss and the calibrated reconstruction loss are 0.0019 and 0.0704, respectively.

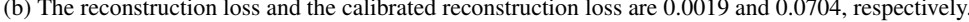
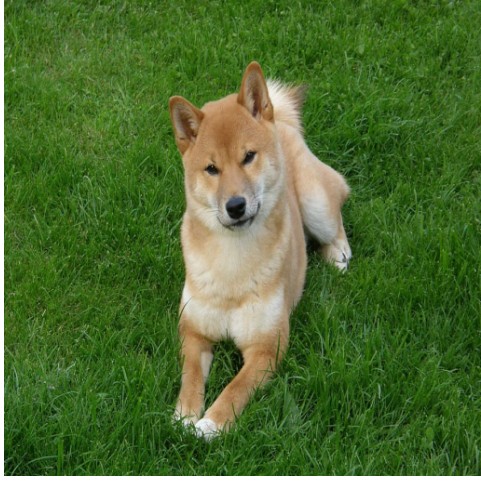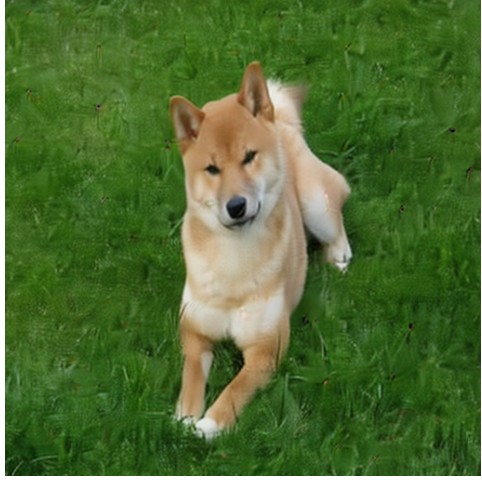

Non-belonging                                    Reverse-engineered

(c) The reconstruction loss and the calibrated reconstruction loss are 0.0028 and 0.2800, respectively.

Fig. 5: More visualization of the non-belonging images for Stable Diffusion v2 [3], and their corresponding reverse-engineered images. The distance metric used here is MSE.

