# OpenReview forum: "Where Did I Come From? Origin Attribution of AI-Generated Images"
_NeurIPS.cc/2023/Conference — NeurIPS 2023 poster_

### Official Review · Reviewer_SUvG · 2023-07-02

**Soundness:** 4 excellent
**Presentation:** 3 good
**Contribution:** 4 excellent
**Rating:** 9
**Confidence:** 4

**Summary:**

This paper addresses the problem of distinguishing between images generated by a generative AI model or acquired by a camera (real images). The motivation comes from the concerns of AI community about the potential misuse and intellectual property (IP) infringement associated with image generation models.

The authors approach the classification problem (generated image vs real image) by first developing an alteration-free and model-agnostic origin attribution method via reverse-engineering (i.e., inverting the input of a particular model for a specific image) followed by computing the reconstruction loss of reverse-engineering to infer the origin. The authors provide the intuition behind the approach by stating that reverse-engineering task is easier for belonging images to a particular model than for non-belonging images generated by some other models


**Strengths:**

The strengths of the paper lie in
(1)	Introducing “alteration-free and model-agnostic origin attribution" algorithm
(2)	Analyzing the differences in the reconstruction loss for reverse engineering between the generated images of a given model and other images, and then checking whether the reconstruction loss of the examined sample falls within the distribution of reconstruction losses observed in the generated images.
(3)	Evaluating the method on eight different image generation models to quantify the accuracy


**Weaknesses:**

The weaknesses of the paper lie in
(1)	Missing description of computational costs
(2)	Reproducibility of the algorithm is the paper presents the algorithm 1 at very high level which makes it difficult to reproduce the algorithm (unless the code becomes available)
(3)	Assuming of having belonging images to a specific model (Section 4.4 refers to 100 images labeled as belonging images to a specific model) which might not always be the case in practice


**Questions:**

Would it be possible to create a table summarizing the combinatorial setups for Architecture = {the same, different} x Training Data = {the same, different, overlapping} given selected two models M1 and M2?

**Limitations:**

The section “Discussion” describes the limitations of the approach in its higher computational cost than the two other approaches based on watermarking [19-22] and classifiers [23-26]. This opens up a question about tradeoffs.

---

> ### Author Rebuttal · Authors · 2023-08-10
>
> Thank you very much for your precious time, thoughtful
> comments, and recognition of the significance of our work.
> We hope the following results and clarifications can
> adequately address your concerns.
>
> **Q1**: Missing description of computational costs.
>
> **A1**: Thanks for your valuable comment. The discussion of
> the efficiency and the computational costs can be found in
> Appendix.E. We will add a more detailed discussion in the
> revised version.
>
> **Q2**: Reproducibility of the algorithm is the paper
> presents the algorithm 1 at very high level which makes it
> difficult to reproduce the algorithm (unless the code
> becomes available).
>
> **A2**: Thank you very much for your insightful suggestion.
> The link to our code repo can be found in line 541 of the
> appendix (supplementary materials).
> We will open-source our code upon acceptance.
>
> **Q3**: Assuming of having belonging images to a specific
> model (Section 4.4 refers to 100 images labeled as belonging
> images to a specific model) which might not always be the
> case in practice.
>
> **A3**: Thanks for your thoughtful comment. In our problem
> formulation, the defender has access to the examined
> model. Thus, the belonging images of it can be generated by
> directly using the examined model. We will add more
> discussion to make it more clear.
>
> **Q4**: Would it be possible to create a table summarizing
> the combinatorial setups for Architecture = {the same,
> different} x Training Data = {the same, different,
> overlapping} given selected two models M1 and M2?
>
> **A4**: Thank you very much for your helpful suggestion. The
> summary of combinatorial setups can be found in Table 3 of
> the attached PDF in the global response. We will add the
> table accordingly in the revised version.

---

> > ### Comment · Reviewer_SUvG · 2023-08-14
> > **Read all reviews. The entire paper rests on Theorem 4.2 and model calibration.**
> >
> > Several reviewers made a comment about missing computational cost. I do not think that have one sentence in Appendix E is addressing the concern of understanding the computational cost: "The average running time for StyleGAN2-ADA and the Consistency Model are 55.16s and 152.83s, respectively. "
> > In my opinion, the computational cost should be mapped to Table 3 and include uncertainties. The authors should also include whether all 64 CPUs and six Quadro RTX 6000 GPUs were utilized during each run.
> >
> > I still think that it is a strong paper among the papers I have reviewed. However, the plethora of reviewers' comments suggest that the narrative should be significantly improved.

---

> > > ### Author Response · Authors · 2023-08-15
> > > **Thank you very much for your feedback and support**
> > >
> > > Thank you very much for your valuable feedback and support. We have started to run the experiments for measuring the computational cost more comprehensively. We will make sure the discussion about the computational cost will be mapped to Table 3 and include uncertainties in our revised version. We will revise our narrative based on the comment accordingly. Thanks again for your insightful comments and support.

---

### Official Review · Reviewer_rv4w · 2023-07-07

**Soundness:** 2 fair
**Presentation:** 3 good
**Contribution:** 3 good
**Rating:** 5
**Confidence:** 4

**Summary:**

This paper develops an origin attribution method to determine whether a specific image is generated by a particular model. The key idea is based on reverse-engineering generation models, and the decision is made by thresholding the reconstruction loss. The proposed method is evaluated on eight different image generation models.

**Strengths:**

1. This method could be applied to different types of generative models (model-agnostic) without requiring any extra operations in the training phase and image generation phase (alteration-free).

**Weaknesses:**

1. The novelty is lacking. The authors state that "this paper is the first work focusing on this problem to infer if a specific sample is generated by a particular model in an alteration-free and model-agnostic manner". However, as far as I know, there are several works working on the same problem listed below, and the proposed methodology is very similar to these works. I doubt whether the authors have conducted sufficient literature research.
[1] Source Generator Attribution via Inversion, CVPRW2019
[2] On Attribution of Deepfakes, arXiv2020

2. The proposed method could not be a "perfect reverse-engineering algorithm". Model inversion has been a difficult problem due to the nonlinearity of neural networks. Existing works make many efforts to improve the precision of inversion. The methods proposed in the work only follow the most simple optimization-based inversion method without innovations in technology.

3.  The experiments are far from sufficient. In the real-world scenario, the number of unknown models is far greater than the known models. However, in Table 3, the author only uses a single model M2 as the negative model. The high accuracy doesn't reflect the performance in the open-world scenario. Besides, there is no comparison with existing methods. The author should at least compare with [1] and [2], which solve the same problem as this paper and use a similar method.

**Questions:**

1. How is the attribution efficiency? As the proposed method is optimization-based,  the attribution procedure may take more time than other methods by straightforward prediction.
2. The authors should conduct more sufficient literature research and conduct more comparison experiments.

**Limitations:**

The limitations are listed in the weaknesses above.

---

> ### Author Rebuttal · Authors · 2023-08-10
>
> Thank you for your time and insightful comments. We have run
> all the suggested experiments. We hope the following new
> clarifications and results can address your concerns. We are
> willing to perform more experiments if you have further
> suggestions.
>
> **Q1**: The authors state that "this paper is the first work
> focusing on this problem to infer if a specific sample is
> generated by a particular model in an alteration-free and
> model-agnostic manner". However, as far as I know, there are
> several works working on the same problem listed below, and
> the proposed methodology is very similar to these works. I
> doubt whether the authors have conducted sufficient
> literature research. Besides, there is no comparison with
> existing methods. The author should at least compare with
> Albright et al. and Zhang et al., which solve the same
> problem as this paper and use a similar method.
>
> **A1**: Thank you for the suggested papers.
>
> * First of all, the problems focused by Albright et al. and
>   Zhang et al. are **different** from our problem. More
>   specifically, their problem is that given an image and a set
>   of provided models, how to determine which model in the given
>   model set is the source of the given image. However, our
>   problem is determining if a given image is generated by
>   a single given model or not, which is a fundamentally
>   different problem. Albright et al. and Zhang et al. have
>   several drawbacks compared to our method. For example,
>   they can not distinguish the real images and the generated
>   images. Also, if the given image is not generated by the
>   model in the provided model set, their method will always
>   give a wrong prediction. Since their problem formulation
>   is different from ours, we did not empirically compare our
>   method to theirs (i.e., their methods are not applicable to our problem).
>
> * Besides the optimization for reverse-engineering, to solve
>   the problem of determining if a given image is generated
>   by a single given model or not, we propose the important
>   calibration step to make the belonging images and
>   non-belonging images more separable. Our origin
>   attribution framework is also based on the designed
>   statistical hypothesis testing. Both these techniques are not discussed in
>   Albright et al. and Zhang et al.
>
> * While Albright et al. and Zhang et al. are limited to the
>   inversion of noise-to-image GANs, our work also includes
>   the reverse-engineering on more models, e.g.,
>   text-to-image diffusion models and GANs. We also discuss
>   different reverse-engineering methods for the latest
>   diffusion models (see Reviewer FZhr A3), which are also
>   insightful to the community. We will add more discussion
>   to make it more clear.
>
> Albright et al., Source Generator Attribution via Inversion. CVPR Workshop 2019.
>
> Zhang et al., On Attribution of Deepfakes. arXiv 2020.
>
> **Q2**: The proposed method could not be a "perfect
> reverse-engineering algorithm". Model inversion has been a
> difficult problem due to the nonlinearity of neural
> networks. Existing works make many efforts to improve the
> precision of inversion. The methods proposed in the work
> only follow the most simple optimization-based inversion
> method without innovations in technology.
>
> **A2**: Thanks for your thoughtful comment. Although the
> reverse-engineering in the real world is not exactly
> perfect, Theorem 4.2 reflects the relaxation and the
> approximations of real-world cases. Thus, Theorem 4.2 is
> meaningful as the guidance of our method. Our experiments in
> Section 5 of the main paper and Table 1 of the
> attached PDF in the global response demonstrates that the
> reconstruction loss values for belonging and non-belonging
> images are highly separable (i,e, the average detection
> accuracy of our method is 96.4%).
>
>
> **Q3**: The experiments are far from sufficient. In the
> real-world scenario, the number of unknown models is far
> greater than the known models. However, in Table 3, the
> author only uses a single model M2 as the negative model.
> The high accuracy doesn't reflect the performance in the
> open-world scenario.
>
> **A3**: Thanks for your helpful comment. We have conducted
> the experiments on 5 negative models accordingly during the
> rebuttal period. The results are shown in the Table 1 of the
> PDF file in the global response. The results demonstrate
> that our method has good performance for distinguishing
> between the belongings of a given model and the images generated
> by other models with different architectures. We will add
> the above results and more discussion in the revised version of
> this paper.
>
> **Q4**: How is the attribution efficiency? As the proposed
> method is optimization-based, the attribution procedure may
> take more time than other methods by straightforward
> prediction.
>
> **A4**: Thank you very much for your thoughtful comment. The
> discussion of the efficiency and the runtime can be found in
> Appendix.E. We admit the computational complexity of our
> method is larger than watermarking and classifiers-based
> method. However, our method is alteration-free and
> model-agnostic while existing methods are not. In addition,
> our method can be accelerated by mixed precision training
> (Micikevicius et al.). We will add more discussion in the
> revised version.
>
> Micikevicius et al., Mixed Precision Training. ICLR 2018.

---

> > ### Author Response · Authors · 2023-08-10
> >
> > **Supplementry for A2**: Although our inversion  approach may appear straightforward, it is underpinned by our theoretical analysis presented in Theorem 4.2. Furthermore, empirical evidence illustrates its remarkable efficacy, as demonstrated in Table 2, Table 3, and Table 4 in our main paper. We also explored various inversion techniques for the state-of-the-art diffusion models, with outcomes suggesting that our approach proves to be the most effective (please refer to Review FZhr-A3 for more details). Notably, our methodology  is general to different types of models including GANs and diffusion models. We hold the conviction that our simple yet highly effective solution for the novel origin attribution problem formulated in Section 3 of our main paper stands to greatly benefit our research field.
> >
> > **Q5**: The authors should conduct more sufficient literature research and conduct more comparison experiments.
> >
> > **A5**: Thank you very much for your constructive suggestions. We will add more discussion about more related literatures (e.g., Albright et al. and Zhang et al.). For the comparisons to Albright et al. and Zhang et al., please refer to A1. Thanks again for your valuable comment.
> >
> > Albright et al., Source Generator Attribution via Inversion. CVPR Workshop 2019.
> >
> > Zhang et al., On Attribution of Deepfakes. arXiv 2020.

---

> > ### Comment · Reviewer_rv4w · 2023-08-16
> > **Thanks for the response**
> >
> > Thanks for the response. The response partly solves my questions. However, some concerns remain unsolved:
> >
> > A1:  The response states that Albright et al. and Zhang et al. address the issue of determining which is the source model of the given image from a provided model set, while this paper aims to determine if a given image is generated by a single given model or not. So the authors argue that the two problems are different. Although the formulation seems different, the former problem is inherently aligned with the latter. To address the task of distinguishing among N models, the attribution process actually involves N iterations of one-versus-rest comparisons. Moreover, as shown in Figure 7, Albright et al. have also conducted one-versus-rest experiments.
> >
> > A4: While inversion-based attribution is inherently more complex than watermarking and classifier-based methods, it would be valuable to incorporate efficiency comparisons with other reverse-engineering methods.
> >
> > A5: Despite slight variations in the experimental configurations of Albright et al. and Zhang et al., their proposed methodologies are also grounded in inversion error and could be easily adapted for this paper's experiments. Consequently, it is suggested to compare the two methods in the future.
> >
> > Based on these concerns, I keep my score.

---

> > > ### Author Response · Authors · 2023-08-18
> > > **Thanks for your feedback (Part 1)**
> > >
> > > Thank you very much for your valuable feedback and
> > > suggestions. Belows are our further responses. We are happy
> > > to answer more questions and perform more experiments if you
> > > have further concerns.
> > >
> > > **Further Response-A1**: Thanks for your helpful feedback.
> > >
> > > * Given an inspected image, Albright et al. and Zhang et al.
> > > works by enumerting and conducting inversion on **all
> > > models** within a set of suspicious candidate models
> > > (referred to as the "candidate set" in this response), and
> > > attribute the model with the lowest reconstruction loss as
> > > the source of the image. Their methods rely on the
> > > assumptions that the inspector can have the **white-box
> > > access to all models** in the candidate set, and the
> > > examined image **must be generated by one of the models in
> > > the candidate set**. These assumptions diminish the
> > > practicability of their methods, whereas **our method does
> > > not have such requirements**.
> > >
> > > * In Table 3 of the main paper, we carry out the
> > >   experiments for distinguishing belonging images of the
> > >   inspected model (i.e., $\mathcal{M}_1$) and images
> > >   generated by other models (i.e., $\mathcal{M}_2$).  It is
> > >   important to clarify that in our settings the inspector
> > >   only has access to the inspected model
> > >   $\mathcal{M}_1$, but he/she does not have any information
> > >   or any access about the model $\mathcal{M}_2$. The goal
> > >   behind these experiments is to investigate our method's
> > >   effectiveness for distinguishing the belonging images of
> > >   the inspected model and the images generated by other
> > >   **unknown** models, which remain undisclosed to the
> > >   inspector. In contrast, both the one-versus-rest
> > >   experiments and other experiments in Albright et al.
> > >   assume the inspector has full white-box access to all
> > >   models involved in the experiments (equivalent to having
> > >   white-box access to both $\mathcal{M}_1$ and
> > >   $\mathcal{M}_2$ in Table 3), and they conduct inversion on
> > >   all models in their candidate set. Thus, our threat model
> > >   and experiment settings are fundamentally different to
> > >   theirs.
> > >
> > > * *"Why will the methods proposed in Albright et al. and
> > > Zhang et al. fail in our formulated problem?":* Our paper
> > > focus on the problem of determining if a given image is
> > > generated by a single given model or not. Consider a
> > > scenario where  a model owner wants to verify if an image is
> > > generated by a model owned by him/her. While our method only
> > > needs to conduct the inversion on this specific model.
> > > Albright et al. and Zhang et al. need to compare
> > > the reconstruction loss of this particular model with a large
> > > number of suspicious models. There are several cases where
> > > Albright et al. and Zhang et al. prove ineffective in
> > > addressing this problem:
> > >
> > > 1. In cases where the inspector lacks white-box access to
> > > some of the suspicious models, deriving reconstruction on
> > >  them and getting inference results becomes infeasible.
> > > Notably, more and more state-of-the-art image generation
> > > models (e.g., such as Midjourney and DALL-E2) are
> > > close-sourced and they only provide the black-box API to the
> > > users.
> > >
> > > 2. Albright et al. and Zhang et al. are prone to making
> > >    wrong predictions if the real source model is not
> > >    included in the candidate set. This is attributed to
> > >    their underlying assumption that the examined image must
> > >    originate from one of the models within the candidate
> > >    set. Ensuring the real source model is included within
> > >    the candidate set is a very hard problem in practice.
> > >
> > > 3. Equally noteworthy, Albright et al. and Zhang et al. do
> > > not work when the inspected images are real images due to
> > > their strong assumption (i.e., the examined image must be
> > > generated by one of the model in the candidate set). Our
> > > method does not have the above problems.
> > >
> > > Will make it more clear in the revised version.

---

> > > ### Author Response · Authors · 2023-08-18
> > > **Thanks for your feedback (Part 2)**
> > >
> > > **Further Response-A4 and A5**:
> > >
> > > * Let's now consider the empirical comparisons. Despite the
> > >   threat models are different, we have conducted the comparison experiments accrodingly. We
> > >   consider the setting for distinguishing the belonging
> > >   images of the inspected model $\mathcal{M}_1$ and the
> > >   generated images of other models $\mathcal{M}_2$, and the
> > >   inspected model (i.e., $\mathcal{M}_1$) here is Stable
> > >   Diffusion 2.0. For our method, we assume the inspector can
> > >   only access $\mathcal{M}_1$. For Albright et al. and Zhang
> > >   et al., we assume the inspector can access both
> > >   $\mathcal{M}_1$ and $\mathcal{M}_2$. The results when
> > >   $\mathcal{M}_2$ is the StyleGAN2-ADA trained on CIFAR-10
> > >   dataset are shown in the following table:
> > >
> > > Method| TP |FP |FN |TN | Acc|
> > > ---- |---- |---- |---- |---- | ---|
> > > Albright et al. | 100  |100 |0 |0 |  50.0%|
> > > Zhang et al. | 100  |100 |0 |0 |  50.0%|
> > > Ours |96|7 |4 |93 | 94.5%|
> > >
> > > The results when $\mathcal{M}_2$ is the Consistency Model trained on ImageNet dataset are shown in the following table:
> > >
> > > Method| TP |FP |FN |TN | Acc|
> > > ---- |---- |---- |---- |---- | ---|
> > > Albright et al. |100  |100 |0 |0 |  50.0%|
> > > Zhang et al. |100  |100 |0 |0 |  50.0%|
> > > Ours |95|10 |5 |90 | 92.5%|
> > >
> > > As evident from the results, our approach demonstrates a
> > > significantly superior performance compared to that of
> > > Albright et al. and Zhang et al. There are several factors
> > > contributing to these outcomes. While Albright et al. and
> > > Zhang et al. attribute the origin of the generated images
> > > solely through direct comparisons of reconstruction losses
> > > across different models, they overlook the variations in
> > > inherent complexities and expressive capabilities among different
> > > models. For example, Stable Diffusion models can easily
> > > achieve relatively low reconstruction losses for both
> > > belonging images and non-belonging images compared to other
> > > simpler models. Consequently, a direct comparison of
> > > reconstruction errors without accounting for the models'
> > > capacities introduces bias. Furthermore, they ignore the
> > > differences in images' inherent complexity (which we address
> > > through our calibration step to mitigate its impact), which
> > > is also a factor that significantly influences performance.
> > > In conclusion, our method outperforms Albright et al. and
> > > Zhang et al. even though they have much stronger
> > > assumptions.
> > >
> > > * Our method is also much more efficient than Albright et
> > >   al. and Zhang et al. when the number of models in their
> > >   candidate sets are large. For example, consider a scenario
> > >   where there are 10000 models in the candidate sets, then
> > >   they need to conduct inversion on 10000 models for
> > >   predicting the source of a single image, which is much
> > >   more time-consuming than our method as we only need to
> > >   conduct inversion on a single model.
> > >
> > > * Please refer to Reviewer FZhr-A3 to see the comparisons
> > >   for different inversion methods (i.e., Song et al., Mokady
> > >   et al., and Parmar et al.) for diffusion models.
> > >
> > >
> > > We will make it more clear in our revised version. Thanks
> > > again for your helpful comment and feedback. We sincerely
> > > hope for your further feedback.
> > >
> > > Albright et al., Source Generator Attribution via Inversion. CVPR Workshop 2019.
> > >
> > > Zhang et al., On Attribution of Deepfakes. arXiv 2020.
> > >
> > > Song et al., Denoising Diffusion Implicit Models. ICLR 2021.
> > >
> > > Mokady et al., Null-text Inversion for Editing Real Images using Guided Diffusion Models. CVPR 2023.
> > >
> > > Parmar et al. Zero-shot Image-to-Image Translation. SIGGRAPH 2023.

---

> > > ### Author Response · Authors · 2023-08-19
> > > **A Friendly Reminder**
> > >
> > > Dear Reviewer rv4w,
> > >
> > > Thanks once again for your valuable comments and precious time. As the discussion period is closing, we genuinely hope you could have a look at the new results and clarifications and kindly let us know if they have addressed your concerns. We would appreciate the opportunity to engage further if needed.
> > >
> > > Best,
> > >
> > > Authors of Paper 5345

---

> > > > ### Comment · Reviewer_rv4w · 2023-08-21
> > > > **Thanks for the response**
> > > >
> > > > Thanks for the feedback. The additional experimental results show that this method is effective in the single-model attribution scenario, which could be hardly achieved by existing multi-model attribution methods like Albright et al. and Zhang et al. Therefore, I increase my score to 4. No further increase is due to the following concern:
> > > >
> > > > While the experimental results provided during the rebuttal phase indicate the effectiveness of the proposed method, a thorough exploration of why effective remains lacking and could hardly be solved in the rebuttal phase. Regardless of the scenario difference and just focus on the technical part, Albright et al., Zhang et al, and this work are all rooted in the reconstruction error of a reverse-engineering process. The principal technical innovation appears to be the incorporation of calibration and hypothesis testing. However, a detailed explanation of why the two parts are essential and sound is absent. Compared to this work, the work in https://arxiv.org/pdf/2306.06210.pdf indeed gives a solid explanation to show that their method is theoretically sound and computationally efficient for single-model attribution. In light of this concern, I keep my score to 4.

---

> > > > > ### Author Response · Authors · 2023-08-21
> > > > > **Thanks for your feedback**
> > > > >
> > > > > Thanks for your valuable feedback and precious time. Below are our further responses.
> > > > >
> > > > > Comment 1: A thorough exploration of why effective remains
> > > > >   lacking. The principal technical innovation appears to be
> > > > >   the incorporation of calibration and hypothesis testing.
> > > > >   However, a detailed explanation of why the two parts are
> > > > >   essential and sound is absent.
> > > > >
> > > > > Response 1: Thanks for your insightful comment.
> > > > >
> > > > > * Our Theorem 4.2 provides a theoretical explanation for high
> > > > >   effectiveness of our reverse-engineering-based method.
> > > > >
> > > > > * For the effects of the calibration step, the reason
> > > > >   why it works is that it can mitigate the influence of the
> > > > > inherent complexities of the inspected image by
> > > > > disentangling the effects of the inherent complexities and
> > > > > the belonging status (See Section 4.2 in our main paper for
> > > > >   more details). Empirically, we have already conducted
> > > > >   the ablation study for it in Section 5.4 of our main paper.
> > > > >   Detailed results can be found in Table 6 of our main
> > > > >   paper. The results demonstrate the calibration step can
> > > > >   significantly improve the detection accuracy of the
> > > > >   proposed method. We will conduct more experiments in a
> > > > >   larger scale and add more results for the ablation study
> > > > >   in our revised version.
> > > > >
> > > > > * The developed hypothesis testing is effective for
> > > > >   determining the inference threshold in our algorithm. Its
> > > > >   effectiveness is grounded on the statistical analysis
> > > > >   performed in Grubbs et al. Compared to other threshold selection methods, it improves the interpretability by providing the significance level, and provides better controls for random variabilities. Hypothesis testing is the standard and objective approach for many decision-making problems such as Li et al. We will add the experiments for comparing the designed
> > > > > hypothesis testing and other threshold selection methods
> > > > > in our revised version.
> > > > >
> > > > > Comment 2: Albright et al., Zhang et al, and this work are all rooted in the reconstruction error of a reverse-engineering process.
> > > > >
> > > > > Response 2: Thank you for your helpful comment. We
> > > > > acknowledge that reconstruction is a commonly-used
> > > > > technique. We provide the theoretical analysis (Theorem 4.2
> > > > > in our main paper) for explaining the effectiveness of it in
> > > > > the specific problem formulated in our paper. Albright et
> > > > > al., Zhang et al do not have such theoretical analysis. We
> > > > > also incorporate the calibration step and the statistical
> > > > > hypothesis testing to address our formulated problem. We
> > > > > will make it more clear in our revised version.
> > > > >
> > > > > Comment 3: Compared to this work, the work in
> > > > > https://arxiv.org/pdf/2306.06210.pdf indeed gives a solid
> > > > > explanation to show that their method is theoretically sound
> > > > > and computationally efficient for single-model attribution.
> > > > >
> > > > > Response 3: Thank you for your suggested paper. This paper
> > > > > is a concurrent work focusing on the single-model
> > > > > attribution problem.
> > > > >
> > > > > * Although the
> > > > > proposed method in https://arxiv.org/pdf/2306.06210.pdf achieves promising performance, it is important to note that  it is
> > > > > **not model-agnostic**. More specifically, it assumes the last
> > > > > layer of the inspected model is invertible. Therefore, it is
> > > > > not applicable to the models that use non-invertible
> > > > > activation functions (such as the commonly-used ReLU
> > > > > function) in the last layer. It also has assumptions on
> > > > > the architectures of the models, making it can not be
> > > > > used for many modern Diffusion Models such as well-known DDPM and
> > > > > Dhariwal et al.
> > > > >
> > > > > * We also want to mention that this paper is posted after
> > > > >   the submission deadline of NeurIPS 2023. We will add the
> > > > >   discussion about this paper in our revised version. Thanks
> > > > >   for your suggestions.
> > > > >
> > > > > Grubbs et al., Sample criteria for testing outlying
> > > > > observations. The Annals of Mathematical Statistics.
> > > > >
> > > > > Li et al., Defending against Model Stealing via Verifying Embedded External Features. AAAI 2022.
> > > > >
> > > > > Ho et al., Denoising Diffusion Probabilistic Models. NeurIPS 2020.
> > > > >
> > > > > Dhariwal et al., Diffusion Models Beat GANs on Image Synthesis. NeurIPS 2021.
> > > > >
> > > > > Thank you once again for your insightful comment and
> > > > > precious time.

---

### Official Review · Reviewer_FZhr · 2023-07-07

**Soundness:** 3 good
**Presentation:** 3 good
**Contribution:** 3 good
**Rating:** 6
**Confidence:** 3

**Summary:**

The method proposes a model-agnostic attribution method. Given a synthesized image as input, the goal is to attribute the correct model that generates the image. Different from prior work, the method is not restricted to a fixed set of models. The main idea of the work is to use reconstruction error -- the model that generates the image should also reconstruct the image better. On top of this, the work introduces a relative reconstruction measure to calibrate the difficulty of reconstruction of each image, along with a thresholding method based on hypothesis testing. The method is tested on multiple generative models or different training datasets, and the method shows strong performance in most cases.

**Strengths:**

1. The author clearly defines the inspector's goal and what the inspector can see. This is helpful in understanding the problem setup easily.

2. The quantitative results look promising, indicating that the relative reconstruction loss is effective for this problem.

3. The method section is well-motivated, where the authors introduce relative reconstruction loss for calibration, and hypothesis testing to find thresholds.

4. Although the method is simple, it is interesting to have a model-agnostic attribution method.

**Weaknesses:**

1. Using reconstruction error is not new for membership inference [1]. Although membership inference focuses on whether an image is used for training the model or not, finding whether a synthesized image is generated by the model shares a similar spirit. It will be good to have more discussion on this.

2. A type of generative model can have multiple ways to reconstruct an input. Currently, the paper does not provide detail about how reconstruction is done for each model. Please check the question sections for this point.

3. Although the authors have mentioned in the limitation section already, the reconstruction-based algorithm requires a huge runtime cost, making it a less favorable option for model attribution.


[1] Hilprecht et al. Monte Carlo and Reconstruction Membership Inference Attacks against Generative Models.

**Questions:**

1. How is reconstruction done on StyleGAN2-ADA and StyleGAN-XL? A common practice is to reconstruct the images by optimizing the extended intermediate latent space (W+) [1], instead of the random noise (z). Will these make a difference in the proposed task?

2. There exist multiple reconstruction strategies for diffusion models as well [2, 3, 4]. It will be great to provide detail on this too.

[1] Abdal et al. Image2StyleGAN++: How to Edit the Embedded Images?
[2] Song et al. Denoising Diffusion Implicit Models.
[3] Mokady et al. Null-text Inversion for Editing Real Images using Guided Diffusion Models.
[4] Parmar et al. Zero-shot Image-to-Image Translation.

**Limitations:**

In my opinion, the authors have addressed the limitations adequately.

---

> ### Author Rebuttal · Authors · 2023-08-10
>
> Thank you very much for your time and insightful comments.
> We hope the following new clarifications and results can
> address your concerns.
>
> **Q1**: Using reconstruction error is not new for membership
> inference. Although membership inference focuses on
> whether an image is used for training the model or not,
> finding whether a synthesized image is generated by the
> model shares a similar spirit. It will be good to have more
> discussion on this.
>
> **A1**: Thank you for your valuable suggestion. We focus on
> the origin attribution problem that inspecting if a given
> image is generated by a given model or not, which is
> different from the membership inference problem that focuses
> on the membership of the training samples. Also, our method
> is guided and supported by our theoretical analysis for the
> origin attribution problem in Theorem 4.2, while the method
> proposed by Abdal et al. is heuristic, and it does not have
> theoretical support for membership inference problem. We
> will add more discussion about the suggested work (i.e.,
> Abdal et al.) accordingly in the revised version.
>
> **Q2**: A type of generative model can have multiple ways to
> reconstruct an input. Currently, the paper does not provide
> detail about how reconstruction is done for each model. How
> is reconstruction done on StyleGAN2-ADA and StyleGAN-XL? A
> common practice is to reconstruct the images by optimizing
> the extended intermediate latent space (W+), instead of
> the random noise (z). Will these make a difference in the
> proposed task?
>
> **A2**: Thanks for your insightful comment. For
> unconditional generative models, we use gradient descent to
> optimize the input in the noise space. For text-to-image
> models, we optimize the input in the intermediate feature
> space. More details about how reconstruction is done for
> each model are provided in Table 2 of the PDF file
> attached to the global response.
>
>
> For StyleGAN2-ADA and StyleGAN-XL, we
> use the random noise space (i.e., z space) to reconstruct
> the images by default. During the rebuttal period, we also
> conducted experiments on using intermediate latent space
> (i.e., W+ space) to reconstruct the images for
> distinguishing belonging images and images generated by
> other models (see setting for the Tabel 3 of the main
> paper). The results on StyleGAN2-ADA with the CIFAR-10 dataset
> are shown in the following table:
>
> Space | Acc|
> ---- | ---|
> z space |  97.0%|
> W+ space| 96.0%|
>
> We also have the results on StyleGAN XL with ImageNet
> dataset and the identical setting used in the Tabel 3 of the main paper, and the results are demonstrated as follows:
>
> Space | Acc|
> ---- | ---|
> z space |  93.0%|
> W+ space| 93.5%|
>
> As can be observed, using the W+ space has
> similar accuracy to using z space, meaning that
> our method is not sensitive to the selection of the input
> spaces used for optimization. We will add more results and
> discussion in the revised version.
>
> **Q3**: There exist multiple reconstruction strategies for
> diffusion models as well. It will be great to provide detail
> on this too.
>
> **A3**: Thank you very much for your constructive comment
> and valuable suggestions. We conduct experiments for
> different inversion methods suggested and our default
> inversion approach on the Stable Diffusion 2 model with the
> setting described in Section 5.3 of the main paper. The
> results are shown in the below table:
>
> Inversion Method | Acc|
> ---- | ---|
> Default |  87.5%|
> DDIM| 55.5%|
> Parmar et al.| 62.5%|
> Mokady et al.| 85.0%|
>
> * DDIM inversion is an inversion approach performing the
>   reverse of the DDIM sampling. It is based on the assumption
>   that the ODE process can be reversed in limited of
>   small steps. It only has unsatisfying inversion performance
>   on the conditional diffusion models (e.g., Stable
>   Diffusion) because it will magnify the accumulated error
>   in the inversion process (since it ignores the
>   classifier-free guidance in the diffusion process). As can
>   be seen in the above table, the accuracy of using DDIM
>   inversion is only 55.5%.
>
> * Parmar et al. improve the DDIM inversion by using an
>   approximated prompt as the conditional guidance in the
>   inversion process. The approximated prompts are generated
>   by a caption model (i.e., BLIP). This method's inversion
>   quality is dependent on the captions used. Given a
>   generated image of the inspected model, since the caption
>   model can not get the accurate prompt used for generating
>   this image, the inversion using this method is also not
>   accurate, leading that the reconstruction losses of the
>   belonging images and non-belonging images are not highly
>   separable (the accuracy is 62.5%).
>
> * Mokady et al. use the diffusion trajectory estimated by
>   the DDIM inversion as the pivotal, and conduct pivotal
>   tuning on the null-text embedding. It achieves comparable
>   accuracy (i.e., 85.0%) to our default inversion method.
>
> We will add more results and discussion in our revised
> version. Thanks again for your valuable suggestions.
>
> **Q4**: Although the authors have mentioned in the
> limitation section already, the reconstruction-based
> algorithm requires a huge runtime cost, making it a less
> favorable option for model attribution.
>
> **A4**: Thank you very much for your thoughtful comment. The
> discussion of the efficiency and the runtime can be found in
> Appendix.E. We admit the computational complexity of our
> method is larger than watermarking and classifiers-based
> method. However, our method is alteration-free and
> model-agnostic while existing methods are not. In addition,
> our method can be accelerated by mixed precision training
> (Micikevicius et al.). We will add more discussion in the
> revised version.
>
> Micikevicius et al., Mixed Precision Training. ICLR 2018.

---

> > ### Comment · Reviewer_FZhr · 2023-08-15
> >
> > I have read the reviews and rebuttal and thank the authors for the clarification. In general, I would still recommend a simple yet effective method to be published in a conference. I would also encourage the authors, in the revised text, to acknowledge that reconstruction is a commonly-used technique in other tasks (e.g., membership inference). In my opinion, adding this reference, along with the clarification on the reconstruction details, will make the paper stronger.

---

> > > ### Author Response · Authors · 2023-08-15
> > > **Thank you very much for your feedback and support**
> > >
> > > Thank you very much for your constructive comment and support. We will recognize accordingly and add the corresponding citations and discussions in our revised version. Thank you again for your valuable feedback and suggestions.

---

### Official Review · Reviewer_FUt3 · 2023-07-07

**Soundness:** 2 fair
**Presentation:** 2 fair
**Contribution:** 2 fair
**Rating:** 5
**Confidence:** 4

**Summary:**

This paper proposes a new problem: given an image $x$ and an generate model $M$, predict whether this image $x$ belongs to $M$ or not. To this end, this paper proposes to do hypothesis testing based on the reconstruction error after conducting latent optimization to reconstruct the given image $x$. They hypothesize that if the image belongs to the model, the reconstruction error while be lower, if it does not belong to, the loss will be higher.

**Strengths:**

- This attribution problem is relative new, despite it has ambiguity with a more popular attribution problem, where given a generated image from a model, how can distribute the credit to each image in the training set.

- their algorithm show high accuracy in their relevant but maybe limited setup.

**Weaknesses:**

while I understand the motivation of considering this problem, but I feel the formulation of this problem is problematic:

- (1) The first is that, this framework will only work if the real images used to train these generative models do not belong to the output space of the generative models. If these training images belong to the output space, then they will be credited as the generated images of the given generative model, which is wrong. If you assume the training images do not belong to the output space, it's also weird, since this is how generative models, such as VAE, is optimized. If a generative model, such as VAE, perfectly fits the training data, your model will wrongly give IP to this generative model.

- (2) relevantly, this framework will determine any reconstructed images as generated images. i.e., given any real image $x$, let's feed it through the vae autoencoder, and then get vae reconstruction as $x'$. Apparently $x'$ belongs to the output space of the vae model, but can we ethically classified it as images generated by the model, and give IP to the VAE? I do not think so.

- (3) what if I train two models with the same architecture on the same dataset (just with different random seeds), then how can you determine which model generate which images? In this case, your method may wrongly attribute output from one model to the other model.

- (4) This method sounds like very easy to get attacked, despite the authors argue that it's advantageous over wartermaking and classifier based one. One apparent question is that, if I do some photoshoping-ish editing or as simple as running a gaussian blurring over the generated images and real images, is this framework still able to distinguish which model generate which image?

- (5) For line 301, you train the same model on different datasets, what if we train different models (architectures) on the same datasets?

- (6) For the study 5.3, one question is that many other papers have shown that diffusion model can almost identically output some images used to train it. So how do you guarantee that there is no similar images in the training set for your generated images?

The calibration step is not very intuitive. Can you use any other model for calibration? will they be equally effective? why do you just choose this consistency model? it's not well justified.

Assuming these generative models are white-box is too strong. The reality is that, many real images are generated by private models where you have no access to even weights, such as Mid-journey.

More realistic situation is also that, let's say we found one image is guilty, then how do we know which model is from? Then it's infeasible to iterate all models to see which model creates this given image.

Writing-wise,

- (1) the abstract is not easy to follow, the inverse engineering makes me think it's something very different that what you actually did, which was just latent optimization for reconstructing the output.
- (2) because of this terminology, it also renders paragraph 67 not easy to understand what you actually do

Isn't Theorem 4.2 trivial? I do not understand why authors put it as a theorem?

**Questions:**

see above

**Limitations:**

see above

---

> ### Author Rebuttal · Authors · 2023-08-10
>
> Thank you for your time and valuable comments. We hope the
> following new clarifications and results can address your
> concerns. We are happy to provide further responses and
> perform more experiments if you have further suggestions.
>
> **Q1**: The first is that, this framework will only work if
> the real images used to train these generative models do not
> belong to the output space of the generative models. If
> these training images belong to the output space, then they
> will be credited as the generated images of the given
> generative model, which is wrong. If you assume the training
> images do not belong to the output space, it's also weird,
> since this is how generative models, such as VAE, is
> optimized. If a generative model, such as VAE, perfectly
> fits the training data, your model will wrongly give IP to
> this generative model.
>
> **A1**: Thank you for your thoughtful comment. We respectively disagree.
>
> * Although the goal of the
> generative model's training is fitting the training data, to the
> best of our knowledge, the training data **can not be perfectly fitted**,
> at least until now. Even the state-of-the-art generative
> models can not reach exactly 0 training loss. Our empirical
> results in Table 2 of the main paper also demonstrate that the training
> samples and the generated samples for the state-of-the-art
> generative models are actually distinguishable by our method.
> There are various existing works (e.g., Rossler et al., Yan
> et al. and Zhu et al.) demonstrate the real images (even the
> training data for the generative models) and the images
> generated by the real-world generative models are
> distinguishable. We want to note that the fundamental goal of
> machine learning is not perfectly memorizing the training
> data. In fact, various existing works (e.g., Carlini et al.)
> demonstrate that deeply memorizing the training data is
> harmful and we should avoid memorizing so much.
>
> * While generative models can produce highly realistic synthetic data, differences between generated and real data often remain detectable. The growing field of generated data detection continues pursuing improvements, suggesting this is still an open research problem rather than a meaningless endeavor.
>
> Carlini et al., Extracting Training Data from Diffusion Models. USENIX Security 2023.
>
> Rossler et al., FaceForensics++: Learning to Detect Manipulated Facial Images. ICCV 2019. (1491 citations)
>
> Yan et al., DeepfakeBench: A Comprehensive Benchmark of Deepfake Detection. arXiv 2023.
>
> Zhu et al., GenImage: A Million-Scale Benchmark for Detecting AI-Generated Image. arXiv 2023.
>
> **Q2**: This framework will determine any
> reconstructed images as generated images.
>
> **A2**: Thanks for your insightful comment. In this paper,
> we focus on the IP infringement and misuse problem related
> to the novel-generated images of the generative models. Note
> that generating creative novel images is also the
> fundamental goal of the image generative models. Forcing the
> generated images to be close to some real images means the
> IP infringement and misuse are mostly associated with the
> imitated real images, instead of the generative model
> itself. The detailed problem formulation, application
> ranges, and use cases of our method can be found in Section
> 3. We admit that the reconstructed images will be considered
> as the belongings of the model, but we do not need to
> consider the attribution of the reconstructed images in our
> use cases. We will make it more clear in the revised
> version.
>
>
> **Q3**: what if I train two models with the same
> architecture on the same dataset (just with different random
> seeds), then how can you determine which model generate
> which images? In this case, your method may wrongly
> attribute output from one model to the other model.
>
> **A3**: Thank you for your constructive comment. For our main use case where the model holder wants to defend the IP of his/her trained models (see Section 3 of our main paper for more details), we focus on
> protecting the IP of the models that are trained on the
> private dataset or close-sourced model architecture. If two
> models trained by different parties only have the difference
> on the random seeds, then both the dataset and the model
> architecture used are open-sourced and we might not need to
> protect the IP of the models. We will add more discussion to
> make it more clear.
>
> **Q4**: Photoshoping-ish editing based adaptive attack.
>
> **A4**: Thank you for your useful comment. The discussion
> about the image-editing-based adaptive attack can be found
> in Appendix.D in the supplementary materials. Following Ali
> et al., to keep the quality of the images while conducting
> editing on them, we use the _1977 Instagram filter to
> conduct image editing. The model used here is the DCGAN
> trained on the CIFAR-10. The results show that the detection
> accuracy of RONAN is 90.5% even under the
> image-editing-based adaptive attack, demonstrating the
> robustness of our method.
>
> Ali et al., ColorFool: Semantic Adversarial Colorization. CVPR 2020.
>
> **Q5**: For line 301, you train the same model on different
> datasets, what if we train different models (architectures)
> on the same datasets?
>
> **A5**: Thanks for your valuable question. The results for
> different models (architectures) on the same datasets can be
> found in Table 2 (also see line 289) in our paper. Also, see
> rv4w-A3 and Tabel 1 of the PDF file in the global response
> for more results.

---

> > ### Comment · Reviewer_FUt3 · 2023-08-14
> > **Thanks for the response**
> >
> > Thank authors for answering my questions. Below are my further comments
> >
> > > A1
> >
> > Thanks for the answer. I know that the point of machine learning is not to overfit, and I was talking about the theoretical flaw of your method. What I was suggesting is a thought experiment. I don't think there is completely no overfitting cases for generative model, and it's just about relative scale: suppose you only have 10 mnist images, very likely you can train a model that is large enough VAE to perfectly fit those pixels, right? More broadly speaking, assuming that real images cannot belong to the output space of generative models is too brute-force.
> >
> > > A2
> >
> > Thank you for your answer, but I don't think it solve my concern at all.
> >
> > > A3 we focus on protecting the IP of the models that are trained on the private dataset or close-sourced model architecture
> >
> > This contradicts to your assumption that you have white-box access to the provided model in line 146.
> >
> > > A4
> >
> > Thank you for sharing this experiment, but I think the given results are far from a comprehensive understanding. What you suggested is your method is robust to editing ( you do not provide how you edit it). Consider a thought experiment, if I run a very large box filter to filter out all high-level details, do you think it's still detectable? As this aspect is very important about the applicability of this method, I recommend conducting a through study understand it.
> >
> > > A5
> >
> > Maybe I am not clear enough. What I meant was, you still train the same method, e.g. VAE or GAN, but you alter the network details, e.g. number of layers, number of neurons.
> >
> > > A6
> >
> > This makes sense and thank you for clarifying it.
> >
> > I feel most of my concerns remain unaddressed, so I will keep my rating.

---

> > > ### Comment · Reviewer_FZhr · 2023-08-15
> > > **Discussion on A1**
> > >
> > > Regarding A1: practically speaking, the focus of attribution should be done on current models that generate photorealistic content, and it has been a while that these models *cannot* fully fit the training set. With that being said, wouldn't an empirical study suffice to show the method's effectiveness? In other words, do we need to handle attribution cases for toy models that overfit to 10 MNIST images in practice?

---

> > > > ### Comment · Reviewer_FUt3 · 2023-08-15
> > > > **Regarding A1**
> > > >
> > > > Hi Reviewer FZhr,
> > > >
> > > > Thank you for sharing your comments. The reason I mentioned 10 MNIST images is just for a thought experiment which illustrates the theoretical flaw of this methods: the author puts a very strong hypothesis that the real image space and the synthetic image space are quite different. However, projecting this hypothesis in the future, I believe the real image space and the synthetic image space will become closer enough.
> > > >
> > > > Putting in to practice (as you mentioned), it's easier to find several scenarios where the hypothesis holds than generally prove it actually holds in various scenarios. Why do I say it this way? The authors only showed it's distinguishable on CIFAR-10/ImageNet trained generative models. For these relative small dataset, the trained generative models hardly generate photo-realistic images (they always have a lot of artifacts which make generated images easily distinguishable even just by eyes). As a result, the Fig 2 results are not surprising.
> > > >
> > > > Given that what really show off the power of generative models are modern diffusion models trained on large scale dataset, a more convincing and sensible experiments could be the following. Find (a) a set of images from Stable Diffusion [here](https://github.com/Stability-AI/stablediffusion) (e.g., some high quality images from prompt hero website); (b) a set of images from DeepFloyd IF [here](https://github.com/deep-floyd/IF); (c) a set of training images for them (e.g., some laion images). Each set doesn't need to be very large, e.g., can be 1k-10k images. Then perform the reconstruction results in Figure 2, and also perform the detection results with the three sets of images, e.g., whether you can detect (b) out from the mixture of (a),(b),and(c).

---

> > > ### Author Response · Authors · 2023-08-15
> > > **Thanks for the feedback (Part 1)**
> > >
> > > Thank you very much for your feedback. Below are our
> > > further responses. We are happy to answer more questions and
> > > perform more experiments if you have further concerns.
> > >
> > > **Further Response-A1**: Thank you very much for your
> > > thoughtful comments and suggestions.
> > >
> > > * As pointed out by existing works (e.g., Frank et al., Wang
> > > et al. and Corvi et al.), morden generative models (e.g.,
> > > VAEs, GANs, and Diffusion models) including the
> > > state-of-the-art models such as Stable Diffusion and DALL-E
> > > will leave forensics traces in the frequency spaces of the
> > > generated images. This is caused by indispensable operations
> > > used in modern generative models (e.g., Upsampling
> > > operations and Convolutional Layers).
> > >
> > > * Also, perfectly fitting the training samples means finding
> > >   the global optimum in the optimization process. However,
> > >   the essential gradient descent optimizers used in modern
> > >   generative models (e.g., SGD and Adam) typically can not
> > >   get the global optimum.
> > >
> > > * Empirically, we conducted the suggested thought
> > > experiments accordingly. In detail, we randomly sample 10
> > > images from MNIST dataset, and use VAE (i.e., Kingma et al.)
> > > with different numbers of neurons in the hidden layer to fit
> > > these 10 images. We set the epoch number to 5000 to ensure
> > > the training losses that measures the distance between the
> > > generated samples and the training samples of the model are
> > > converged. The detailed final training losses with different
> > > model sizes are shown as follows:
> > >
> > > Num of Neurons in Hidden Layer | Training Loss|
> > > ---- | ---|
> > > 1 |  184.24|
> > > 5| 105.27|
> > > 10| 77.20|
> > > 50| 53.52|
> > > 100| 53.23|
> > > 500| 53.04|
> > > 1000| 53.81|
> > >
> > > With the increases in the model sizes, the final training
> > > losses reduce at first. However, when the model is large
> > > enough, the final training loss becomes stable at the region
> > > from 53.00-54.00. The results mean that even the VAEs with
> > > enough large sizes can not fit all pixels perfectly.
> > > We also conduct the reverse-engineering of the training
> > > samples on the trained VAE with 1000 neurons in the hidden
> > > layer. The results show that we can not reconstruct the
> > > exact training samples.
> > >
> > > Based on the above analysis and empirical results, at least we can conclude that the probabilities that the real images belong to the output space of generative models are very low.
> > >
> > > Kingma et al., Auto-Encoding Variational Bayes. ICLR 2014.
> > >
> > > Frank et al., Leveraging Frequency Analysis for Deep Fake Image Recognition. ICML 2020.
> > >
> > > Wang et al., CNN-generated Images Are Surprisingly Easy to Spot... for Now. CVPR 2020.
> > >
> > > Corvi et al., On the Detection of Synthetic Images Generated by Diffusion Models. arXiv 2022.
> > >
> > > **Further Response-A2**: Thanks for your insightful
> > > feedback.
> > >
> > > * While we admit that the reconstructed images will be
> > > considered as the belongings of the model, it is essential
> > > to clarify that infering an image as a belonging of a model
> > > does not imply the IP of this image is totally belonging to
> > > this model. In fact, determining the ownership of the IP
> > > related to the generated images remains an unresolved
> > > challenge in the field of law. This complexity arises due to
> > > the involvement of multiple entities (such as contributors
> > > of training data, model trainers, input/prompt providers,
> > > and the models themselves) throughout the image generation
> > > process. The infering results of our origin attribution
> > > method can serve as a valuable reference for addressing IP
> > > protection concerns, instead of a definitive conclusion.
> > >
> > > * In this paper, we focus on our formulated origin
> > > attribution problem of the generative models. Beyond serving
> > > as a reference for safeguarding intellectual property, our
> > > method has versatile applications, including tracing the
> > >  source of maliciously generated images and detecting
> > > AI-powered plagiarism. For instance, imagine a scenario
> > > where an individual generates AI-created images (e.g., using
> > > Midjourney) and dishonestly presents them as their own
> > > original artwork (e.g., photographs and paintings) to gain
> > > recognition and reputation. In such cases, the model owner
> > > (e.g., Midjourney's owner) may suspect that the image is
> > > generated using their model (e.g., Midjourney). Our proposed
> > > method can then be employed to uncover instances of
> > > AI-powered plagiarism. Importantly, the concern regarding the
> > > impact of reconstructed images is minimized in this context.
> > > This is because the malicious user's goal is garnering
> > > acclaim through the dissemination of novel images, and they
> > > are unlikely to use the reconstructed versions of real
> > > images for this purpose.
> > >
> > > Thanks again for your thoughtful comment, we will revise our paper accordingly to make it more clear.

---

> > > > ### Comment · Reviewer_FUt3 · 2023-08-15
> > > >
> > > > Thank you for the further response.
> > > >
> > > > > This is caused by indispensable operations used in modern generative models (e.g., Upsampling operations and Convolutional Layers).
> > > >
> > > > agreed
> > > >
> > > > > perfectly fitting the training samples means finding the global optimum in the optimization process. However, the essential gradient descent optimizers used in modern generative models (e.g., SGD and Adam) typically can not get the global optimum.
> > > >
> > > > Maybe I am wrong, but plenty of theory papers are showing global optimum is achievable with overparameterized network. This is why I mentioned the thought experiment.
> > > >
> > > > > The detailed final training losses with different model sizes are shown as follows
> > > >
> > > > Does the training loss here you shown include the kl-divergence part? or just the pixel reconstruction loss by running inference mode on the training images (e.g., use the mean predicted by the encoder to reconstruct the image)?
> > > >
> > > > In light of the response, I increase my score to 3.
> > > >
> > > > But if you can deliver convincing detection results between: (1) stable diffusion images; (2) deepfloyd-if images; and (3) real images. I am willing to consider further increase.

---

> > > > > ### Author Response · Authors · 2023-08-16
> > > > > **Thanks again for your feedback**
> > > > >
> > > > > Thank you very much for your thoughtful feedback.
> > > > >
> > > > > * We are aware of the theoretical studies (such as Chizat et
> > > > >   al., Du et al., and Fang et al.)  about the global
> > > > >   convergence of the overparameterized network. We want to
> > > > >   mention that these theoretical investigations revolve
> > > > >   around classification models. It's important to note that
> > > > >   the loss function of the modern generative models (e.g.,
> > > > >   GANs, VAEs and Diffusion models) exhibit a greater level
> > > > >   of complexity compared to classifiers. For example, the
> > > > >   training of the GANs encompasses iterative updates
> > > > >   involving both generators and discriminators. Furthermore,
> > > > >   these theoretical analyses are constrained to specific
> > > > >   model architectures, such as fully connected networks.
> > > > >   Consequently, their theoretical analysis on the simpler
> > > > >   classifiers may not accurately reflect the conditions
> > > > >   inherent in modern generative models.
> > > > >
> > > > >   Chizat et al., On the Global Convergence of Gradient Descent for Over-parameterized Models using Optimal Transport. NeurIPS 2018.
> > > > >
> > > > >   Du et al., Gradient Descent Provably Optimizes Over-parameterized Neural Networks. ICLR 2019.
> > > > >
> > > > >   Fang et al., Over Parameterized Two-level Neural Networks Can Learn Near Optimal Feature Representations. arXiv 2019.
> > > > >
> > > > > * The reported training losses are total losses include the
> > > > > kl-divergence part. There are two parts in the total
> > > > > training loss *$\mathcal{L}_{total}$*: the reconstruction part
> > > > > *$\mathcal{L}_{reconstruction}$* and the kl-divergence part
> > > > > *$\mathcal{L}_{divergence}$*, i.e., *$\mathcal{L}_{total}$* = *$\mathcal{L}_{reconstruction}$*+*$\mathcal{L}_{divergence}$* (See
> > > > > Equation 10 in Kingma et al.). We show the detailed value for
> > > > > each part in the following table:
> > > > >
> > > > > Num of Neurons in Hidden Layer | $\mathcal{L}_{total}$|$\mathcal{L}_{reconstruction}$|$\mathcal{L}_{divergence}$|
> > > > > ---- | ---|---|---|
> > > > > 1 |  184.24|183.93|0.31|
> > > > > 5| 105.27|99.68|5.59|
> > > > > 10| 77.20|69.07|8.13|
> > > > > 50| 53.52|47.55|5.97|
> > > > > 100| 53.23|47.99|5.24|
> > > > > 500| 53.04|48.21|4.83|
> > > > > 1000| 53.81|48.93|4.88|
> > > > >
> > > > > Kingma et al., Auto-Encoding Variational Bayes. ICLR 2014.
> > > > >
> > > > > We will make it more clear in our revised version.
> > > > >
> > > > > * We have already started to conduct the suggested
> > > > > experiemnt on stable diffusion images, deepfloyd-if images,
> > > > > and real images. We will keep you updated once we have the
> > > > > results. Thank you very much for your constructive
> > > > > suggestions for this valuable experiment.
> > > > >
> > > > > Thanks again for your insightful comments and suggestions,
> > > > > which have improved our submission.

---

> > > > > ### Author Response · Authors · 2023-08-19
> > > > > **Results on the suggested experiments**
> > > > >
> > > > > Dear Reviewer FUt3,
> > > > >
> > > > > Thank you very much for your thoughtful comment and
> > > > > feedback. Below are the results for the suggested
> > > > > experiments on stable diffusion images, deepfloyd-if images,
> > > > > and real images. We are happy to answer more questions and
> > > > > perform more experiments if you have further concerns.
> > > > >
> > > > > In our experiments, we use the Stable Diffusion 2 model as
> > > > > the inspected model, and we conducted two corresponding
> > > > > experiments accordingly: (1) Distinguishing 1000 images
> > > > > generated by stable diffusion model and 1000 images
> > > > > generated by DeepFloyd-IF model. (2) Distinguishing 1000
> > > > > images generated by stable diffusion model and 1000 randomly
> > > > > sampled images in Laion dataset. The prompts for the
> > > > > generated images is randomly sampled on the PromptHero
> > > > > website as suggested.
> > > > >
> > > > > The results for distinguishing stable diffusion images and
> > > > > deepfloyd-if images are shown in the follwing table:
> > > > >
> > > > >  TP |FP |FN |TN | Acc|
> > > > > ---- |---- |---- |---- | ---|
> > > > >  957  |124 |43 |876 |  91.6%|
> > > > >
> > > > > The results for distinguishing stable diffusion images and
> > > > > Laion images are demonstrated in the follwing table:
> > > > >
> > > > >  TP |FP |FN |TN | Acc|
> > > > > ---- |---- |---- |---- | ---|
> > > > >  961  |188 |39 |812 |  88.7%|
> > > > >
> > > > > As can be observed, our method can effectively distingushing
> > > > > the stable diffusion images and deepfloyd-if images (i.e.,
> > > > > the detection accuracy is 91.6%). It also achieves high
> > > > > detection accuracy (i.e., 88.7%) on distingushing the stable
> > > > > diffusion images and the Laion images. We will add more
> > > > > results and discussions in our revised version.
> > > > >
> > > > > We sincerely thank you again for your constructive suggestions
> > > > > and feedbacks.
> > > > >
> > > > > Bests,
> > > > >
> > > > > Authors of paper 5345

---

> > > > > > ### Comment · Reviewer_FUt3 · 2023-08-21
> > > > > >
> > > > > > Thank you for your efforts on clarifying my questions, rating adjusted accordingly.

---

> > > > > > > ### Author Response · Authors · 2023-08-21
> > > > > > >
> > > > > > > Thank you very much for your feedback. We sincerely appreciate your thoughtful comments and valuable suggestions, which have improved our submission.

---

> > > ### Author Response · Authors · 2023-08-15
> > > **Thanks for the feedback (Part 2)**
> > >
> > >
> > > **Further Response-A3**: Thanks for your valuable feedback.
> > >
> > > * We want to clarify that the main user of our method is the
> > > model owners. For example, a model owner trains a model
> > > using his/her private dataset or his/her own close-sourced
> > > model architecture (and only provide the black-box API to
> > > the downstream users of this model, such as Midjourney and
> > > DALL-E 2), and they can use our method to infer or
> > > demonstrate if a specific image is generated by this model.
> > > Since the model belongs to the model owner, it is natural
> > > that the model owner has white-box access to the model.
> > > Thus, having white-box access and using private datasets or
> > > close-sourced model architecture is not contradictory.
> > >
> > > * Furthermore, it is important to highlight that as modern
> > > models continue to develop, their sizes are progressively
> > > expanding, resulting in the demand for substantial time and
> > > resources to train these larger models. For example, the
> > > training of GPT-4 incurred a cost exceeding $100 million
> > > (source: https://en.wikipedia.org/wiki/GPT-4). It is
> > > noteworthy that, in practice, industries are unlikely to
> > > undertake repeated training of such state-of-the-art models
> > > solely varying in random seeds.
> > >
> > > We will make it more clear in our revised version.
> > >
> > > **Further Response-A4**: For the experiments in the Section
> > > Appendix.F, the details of the _1977 Instagram filter we
> > > used can be found in the link we provided in line 597 of the
> > > Appendix (supplementary material). We also conducted the
> > > suggested thought experiment accordingly. The image editing
> > > method is the suggested image box filter. The other settings
> > > are identical to that in Section Appendix.F. Besides the
> > > detection accuracy (Acc) of our method, we also demonstrate
> > > the Structural Similarity Index (i.e., SSIM proposed in Wang et al.) between the original images and
> > > the edited images, which can measure the similarity between
> > > them. A higher SSIM value means the edited images are more
> > > similar to the original images. The results under different
> > > box sizes of the image filter are shown in the following
> > > table.
> > >
> > > Box Size | Acc| SSIM|
> > > ---- | ---|---|
> > > 1 |  92.5% |0.8920|
> > > 2 |  83.0%|0.7446|
> > > 3 |  58.0%|0.5174|
> > > 4 |  53.5%|0.3530|
> > >
> > > As can be observed, our method remains effective under
> > > relatively small box sizes. As the box sizes expand,
> > > however, the detection accuracy of our method diminishes.
> > > This outcome is understandable and acceptable, as it
> > > corresponds to a rapid reduction in the Structural
> > > Similarity Index (SSIM) between the edited images and their
> > > unaltered counterparts. When employing larger box sizes, it
> > > is conceivable that an adaptive attacker might find ways to
> > > elude our method's scrutiny, yet this comes at the cost of
> > > substantially compromising the quality of the edited images.
> > > Consequently, our method maintains its effectiveness even in
> > > the face of adaptive attacks that seek to maintain the
> > > quality of the edited images.
> > >
> > > Wang et al., Image quality assessment: from error visibility to structural similarity. TIP 2004.
> > >
> > > **Further Response-A5**:
> > > Thanks for your constructive suggestions. We conducted the experiments as suggested. The model and the dataset used here are DCGAN and CIFAR-10, respectively. We first provide the empirical results when the $\mathcal{M}_1$ (i.e., the inspected generator) and the $\mathcal{M}_2$ (i.e., the other generator) have different numbers of layers. The results are shown in the following table:
> > >
> > > $\mathcal{M}_1$'s Number of Layers|$\mathcal{M}_2$'s Number of Layers| Acc|
> > > ---- |---- | ---|
> > > 4 |2 |  97.5%|
> > > 2 |4| 98.0%|
> > >
> > > We also demonstrate the results when the $\mathcal{M}_1$ (i.e., the inspected generator) and the $\mathcal{M}_2$ (i.e., the other generator) have different numbers of channels in the first Convolutional layer in the following table:
> > >
> > > $\mathcal{M}_1$'s Number of Channels in the first Conv Layer |$\mathcal{M}_2$'s  Number of Channels in the first Conv Layer | Acc|
> > > ---- |---- | ---|
> > > 64 |48 |  97.5%|
> > > 48 |64| 96.0%|
> > >
> > > As can be observed, our method achieves high detection accuracy among these settings.
> > > These results indicate our method is effective when
> > > $\mathcal{M}_1$ and $\mathcal{M}_2$ are trained using the same method, but with different network details, e.g. number of layers, and number of neurons.

---

### Author Rebuttal · Authors · 2023-08-10


We sincerely thank all reviewers for their thoughtful
comments and precious time. We provide our responses below
to address concerns. Please let us know if there is anything
still not clear. We are willing to answer more questions and
perform more experiments if the reviewers have further
concerns.

Due to the length limition, we put the responses for Q6, Q7, Q8, Q9 and Q10 of Review FUt3 in here.

**Review FUt3-Q6**: For the study 5.3, one question is that many other
papers have shown that diffusion model can almost
identically output some images used to train it. So how do
you guarantee that there are no similar images in the
training set for your generated images?

**Review FUt3-A6**: Thanks for your constructive question. We are aware
of some methods (e.g., Carlini et al. and Webster et al.)
that can extract generated images similar to some training
samples of the model. Although the extracted images are much
more similar to some training samples than the randomly
generated images, these generated images still have a
certain $l_2$ distance to the corresponding training images,
and there are obvious artifacts for human vision in these
generated images (see Carlini et al.). Thus, the
distribution of the extracted images is different from that
of the corresponding training images. We also conduct
experiments on using our method to distinguish the extracted
images (by using Webster et al.) and their corresponding
training images. The model used is Stable Diffusion. The
results are that our method still achieves 85.0% accuracy
for distinguishing the memorized training samples and the
corresponding generated samples. We will add more details
and results in our revised version.

Carlini et al., Extracting Training Data from Diffusion Models. USENIX Security 2023.

Webster et al., A Reproducible Extraction of Training Images from Diffusion Models. arXiv 2023.


**Review FUt3-Q7**: Calibration step and reference models.

**Review FUt3-A7**: Thank you very much for your constructive comment.
We have conducted experiments that used different models as
reference models during the rebuttal. The inspected model
here is the StyleGAN2-ADA model trained on CIFAR-10 dataset
and the setting here is identical to that used in the Table
2 of the main paper (belongings vs training data)
distinguishing the belonging images and the results can
be found in the following table.

Reference Model | Acc|
---- | ---|
Consistency Model |  97.0%|
StyleGAN XL| 95.0%|
Stable Diffusion| 96.0%|

As can be observed, using different reference models yields
similar results, meaning that our method is not sensitive to
the selection of the reference models. We will add more
discussion and results in our revised version.

**Review FUt3-Q8**: Assuming these generative models are white-box is
too strong.

**Review FUt3-A8**: Thanks for the useful comment. The detailed
application ranges and use cases of our method can be found
in Section 3. The inspectors in all use cases can have
white-box access to the model, and they know the range of
the examined models (i.e., they do not need to iterate all
models). For example, the main use case of our method is
protecting the IP of the model owner. In this scenario, a
party suspects that a specific image may have been generated
by their generative model without authorization, such as if
a malicious user has stolen the model and used it to
generate images. The party can then request an inspector to
use our proposed method to infer if the doubtful image was
indeed generated by their particular model. The situation
where the inspector does not have white-box access to the
model and needs to iterate all models is out of the scope of
this paper. We will add more discussion to make it more
clear in our revised version.

**Review FUt3-Q9**: Theorem 4.2.

**Review FUt3-A9**: Although it is straightforward, Theorem 4.2
establishes the theoretical separability of the
reconstruction loss values for belonging and non-belonging
images. Thus, Theorem 4.2 is meaningful as the guidance of
our method.

**Review FUt3-Q10**: Other writing issues.

**Review FUt3-A10**: Thank you very much for your helpful suggestions.
We will revise accordingly in the revised version.

---

### Decision · Program_Chairs · 2023-09-21

**Decision:**

Accept (poster)

**Comment:**

This submission presented a model-agnostic attribution method by introducing a relative reconstruction measure to calibrate the difficulty of reconstruction of each data. Overall, the reviewers are positive and appreciate the idea and contribution of this work. It is of good value compared to previous works in terms of three aspects: (1) they provide a theory prove about why reconstruction loss can tell the belonging (though intuitively it is not surprised), (2) they contribute the calibrated reconstruction loss and hypothesis testing (two points mentioned by you as well) and support with ablation and explanation (but can be improved), (3) they analyze the attribution in diffusion models, which is the latest mainstream model in content generation but the copyright issue is frequently raised and urgently needs some solutions. After a fruitful discussion and checking all reviews, a decision of acceptance is made. There are still some concerns, for example, the literature review is not deep, as two closely related works in the above discussion are totally missed. Discussion with them should be added through revision.